# Development of MPFC function mediates shifts in self-protective behavior provoked by social feedback

Leehyun Yoon[1], Leah H. Somerville[2] & Hackjin Kim [1]

How do people protect themselves in response to negative social feedback from others? How does such a self-protective system develop and affect social decisions? Here, using a novel reciprocal artwork evaluation task, we demonstrate that youths show self-protective bias based on current negative social evaluation, whereas into early adulthood, individuals show self-protective bias based on accumulated evidence of negative social evaluation. While the ventromedial prefrontal cortex (VMPFC) mediates self-defensive behavior based on both current and accumulated feedback, the rostromedial prefrontal cortex (RMPFC) exclusively mediates self-defensive behavior based on longer feedback history. Further analysis using a reinforcement learning model suggests that RMPFC extending into VMPFC, together with posterior parietal cortex (PPC), contribute to age-related increases in self-protection bias with deep feedback integration by computing the discrepancy between current feedback and previously estimated value of self-protection. These findings indicate that the development of RMPFC function is critical for sophisticated self-protective decisions.

[1] Department of Psychology, Korea University, 145 Anam-ro, Seongbuk-gu, Seoul 136-701, Republic of Korea. [2] Department of Psychology and Center for Brain Science, Harvard University, Cambridge, MA 02138, USA. Correspondence and requests for materials should be addressed to H.K. (email: hackjinkim@korea.ac.kr)

Numerous studies have demonstrated that adults maintain positive self-views in spite of a constant stream of others' evaluations[1,2]. For example, people adopt a number of subtle and sophisticated strategies to deflect negative social feedback by derogating the evaluator[3], exaggerating one's desirability in comparison to the average other[4], or even exerting displaced aggression toward innocent others[5]. In general, these results imply that in the face of a constant stream of positive and negative social evaluations, adults incorporate evaluative information and use it to guide social interactions in ways that are not necessarily faithful to the underlying statistics of the social environment. Rather, they could select social behavior in ways that are biased toward preserving self-views.

Given the key links between adolescent development, social rejection, and risk for mental illness[6,7], it is crucial to consider self-protective biases from a developmental perspective. Emerging research suggests that the self-protective biases exhibited by adults may be lacking in adolescents, leaving them vulnerable to drops in self-esteem following social feedback[8]. However, another study in children[9] and adults[10] used the same task and showed that both age groups engaged in self-protection by expressing aggression to those who gave negative evaluation, although no direct comparison was made between age groups. Much more work is needed to understand how self-protective biases emerge across development, an objective of the present study which included 10- to 25-year-old participants.

Although self-protective behaviors have been extensively investigated in a wide range of laboratory-based studies[2,4], many of the studies have focused on immediate response to a single instance of social feedback, while a few studies[5] have characterized self-protection triggered by general sense of being rejected/accepted accumulated over a long time horizon. In addition, simultaneous measurement of immediate and accumulated feedback effects has rarely been done in a repeated-measures design. The present study is designed to examine relative impact of immediate and accumulated feedback on other evaluation, defined as self-protective behavior, in a single participant, focusing on age-related differences in such behaviors. We predicted that, comparing individuals from late childhood to early adulthood, younger individuals would utilize immediate feedback to trigger self-protection, whereas increasing age would predict a shift in reliance on accumulated, rather than immediate, feedback. Supporting this hypothesis, it was shown that those in late childhood and early adolescents display highly variable state self-esteem responding to immediate context[11,12], whereas adults show more stable state self-esteem[13], and build self-concepts based on abundant past experiences[14–17]. Moreover, other studies have reported that, throughout adolescence, people develop more sophisticated knowledge of strategies for dealing with interpersonal conflicts[18], and shift from simple/direct to strategic/indirect tactics for expressing aggression[5,19,20].

The present study uses a cross-sectional design to investigate the underlying neural mediators of such an age-related shift in self-protective behavior. We had an a priori prediction that the medial prefrontal cortex would mediate age-related changes in feedback-related self-protection, because the ventromedial prefrontal cortex (VMPFC) is well-known for its role in self-protection against social evaluation[4,21]. Importantly, this region encodes social-feedback prediction error (PE) signals contributing to trial-by-trial changes in self-worth[22], and the degree to which this region encodes the valence of social feedback was associated with individual differences in self-esteem[23], a psychological construct closely related to the sensitivity to social evaluation[24].

Other recent neuroimaging studies indicated that the rostromedial prefrontal cortex (RMPFC), located slightly dorsal to VMPFC, may contribute to self-protection due to feedback history. For example, this region computes the discrepancy between experienced and expected social outcomes, which is then used to prompt self-protecting decisions[25] or self-protecting cognitive distortions[26]. In addition, RMPFC was found to encode accumulated reward outcomes on a trial-by-trial basis during affective decision-making[27], and maturation of the RMPFC function seems to be critical for integrating social information across time and for coordinating a temporally extended representation of such information[28].

In the present study, we developed a novel reciprocal artwork evaluation task designed to capture a subtle fluctuation of trial-by-trial self-protective decision bias due to social feedback, as measured by negative evaluation of the partner. We focused on the creativity of one's artwork as a target of social evaluation because creativity is a highly valued trait[29], yet is also somewhat subjective which allows room for a wide range of opinions from different observers[30]. An additional benefit of the present task is the capacity to apply formal computational models to compute how shallow or deep of feedback history is being integrated into triggering self-protective behaviors.

Sixty participants aged from 10 to 25 visited laboratory twice, 2 weeks apart. At the first visit, they were instructed to make a creative artwork (Fig. 1b) utilizing everyday materials (Fig. 1a) whose creativity was to be evaluated by other similar-aged participants. At the second visit, participants performed a reciprocal artwork evaluation task with 75 partners. In each trial, participants watched the partner's feedback on their artwork and then evaluated the partner's artwork (Fig. 1c). The behavioral results show that immediate self-protective behavior after current feedback is most apparent in late childhood/early adolescence, showing age-related decreases, whereas into adulthood, people monitor how they have been generally evaluated by previously encountered individuals and engage in self-protection when the accumulated evidence indicates their negative social standing. Neuroimaging analysis demonstrates that, while VMPFC mediates self-protective behavior in general, RMPFC function is engaged for the emergence of self-protective behavior by integrating large amounts of previous social feedback. This study expands our understanding of how neurodevelopment shapes social-feedback monitoring and self-protective behaviors.

## Results

**Behavioral results**. A logistic regression model (Eq. (1)) was fitted to individuals' decision data to estimate the degree to which their evaluation of partners' artworks was influenced by the immediate current feedback (cFB) they had received from the same individual (i.e., cFB influence), and/or the accumulated feedbacks (accFBs) they had received thus far from all previous raters (i.e., accFB influence) (see Fig. 2a), controlling for the objective creativity (OC) of each trial. All statistical tests were two-tailed.

Consistent with our prediction, we found that the younger were more likely to match their ratings of a partner's artwork with the feedback they had just received from the partner, and that such an current feedback effect decreased linearly with age (cFB influence: linear regression, $F(1, 56) = 5.420$, $p = 0.02$). However, the tendency for cumulative ratings (integrated across all previous trials) to influence participants' evaluations showed a linear increase as a function of age, with greater accumulated negative ratings predicting a greater tendency to rate the partner's artwork negatively (accFB influence: linear regression, $F(1, 56) = 8.434$, $p = 0.005$). When the age-related dissociation between cFB and accFB influence was further tested, there was a robust interaction between age and feedback type (2: cFB, accFB) (ANCOVA, $F(1, 56) = 19.234$, $p < 0.001$; Fig. 2b), indicating an

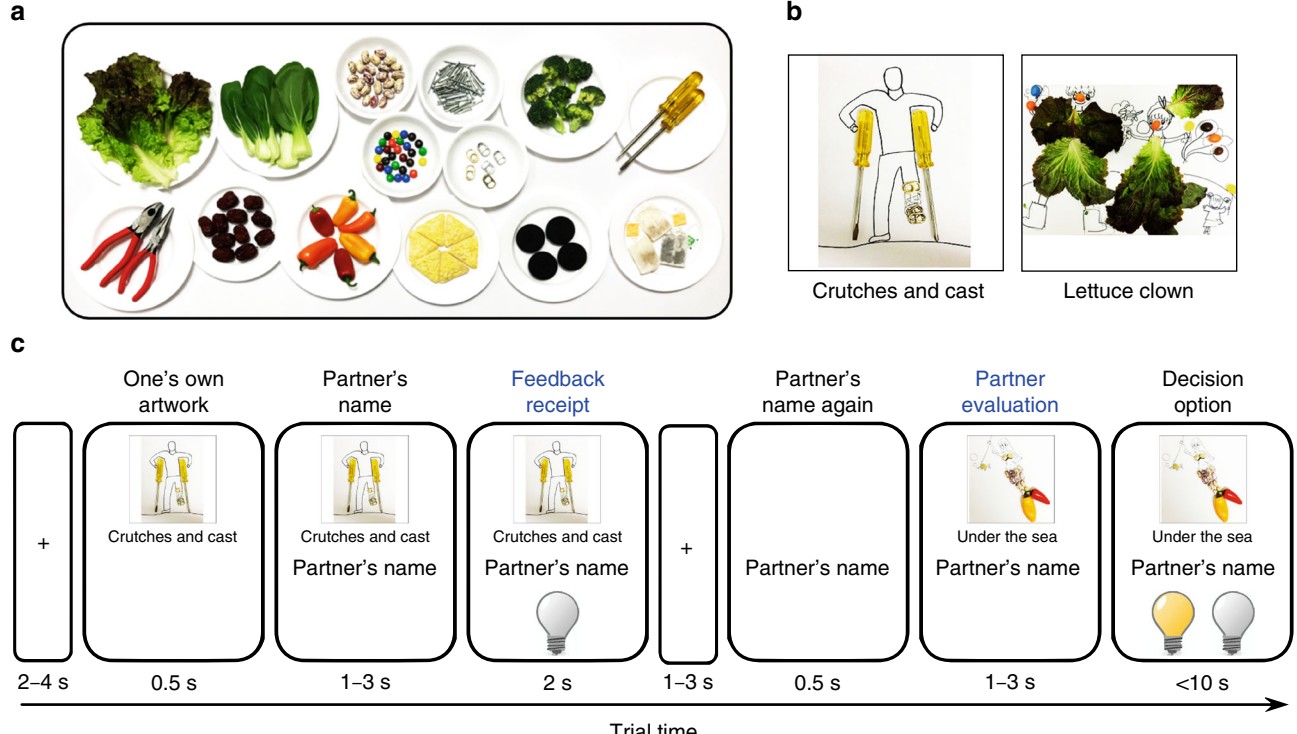

**Fig. 1** Creative artwork (first visit) and reciprocal artwork evaluation (second visit) task. **a** Everyday materials provided for making a creative artwork (first visit). Participants were instructed to make a creative artwork by using one or two from the provided set of everyday materials and label its name. **b** Example artworks from an adult (left) and a child (right). **c** The sequence of a typical trial of the reciprocal evaluation task (second visit). In each trial, a participant saw his/her own artwork, followed by the partner's name, and then viewed the partner's feedback on his/her artwork. A lighted and unlighted bulb indicates "creative" and "not creative" evaluation feedback, respectively, from the partner (see Method for presentation of neutral feedback). Next, they saw the partner's name again, and evaluated the partner's artwork, and then made a decision about the creativity of the artwork

opposite age effects of cFB and accFB influence. When the accFB influence was once again tested in an independent sample with adults only ($N = 28$), we found a successful replication of a significant accFB influence (one-sample $t$ test, $t(27) = 2.874$, $p = 0.008$). This suggests that adults (or older youth) engage self-protection by derogating others when accumulated social evaluations turn negative, unlike younger youths who immediately derogate others who gave a negative feedback.

**Neural mediator of cFB influence.** To identify brain regions linked to the cFB influence, we estimated individuals' cFB parametric maps at the feedback receipt event and the partner-evaluation event (GLM1) and the resulting maps were regressed against corresponding individuals' behavioral cFB parameter estimates, controlling for behavioral parameter estimates of accFB and OC.

The resulting statistical maps identified VMPFC (Brodmann Area 11; 23 voxels at $x = 10$, $y = 24$, $z = -18$; small volume correction family-wise error (SVC FWE) $p < 0.05$, Fig. 3a and Supplementary Fig. 2a; see Supplementary Fig. 1a for anatomical region of interest (ROI) mask), indicating the degree to which VMPFC activity negatively correlated with the cFB value (i.e., greater activation to negative cFB value) at the feedback receipt event predicted individual differences in the cFB influence on subsequent ratings of partners' artworks. A mediation analysis showed that the degree to which VMPFC signals strongly encode cFB value significantly mediated age-related decreases in the cFB influence (i.e., indirect effect: $B = -0.02$, SE = 0.01, 95% CI: [−0.0382 to −0.0036]; Fig. 3c), suggesting that the VMPFC activity differentiating cFB varying from negative to positive

values is a key mediator of the age-related decrease in self-protecting behavior based on the cFB.

**Neural mediator of accFB influence.** To localize brain regions linked to the accFB influence, we estimated individuals' accFB parametric maps at the feedback receipt event and the partner-evaluation event (GLM1) and the resulting maps were regressed against corresponding individuals' behavioral accFB parameter estimates, controlling for behavioral parameter estimates of cFB and OC. Results indicated that VMPFC (BA11; 23 voxels at $x = -6$, $y = 36$, $z = -14$; SVC FWE $p < 0.05$) and RMPFC (BA10; 194 voxels at $x = -4$, $y = 62$, $z = 16$; SVC FWE $p < 0.05$) (Fig. 3b and Supplementary Fig. 2b; see Supplementary Fig. 1b for anatomical ROI mask) signals negatively correlating with the accFB value (i.e., greater activation to more negative accFB value) at the partner-evaluation event predicted individual differences in the accFB influence. Moreover, exploratory whole-brain analyses also identified a large RMPFC cluster (BA10; 197 voxels at $x = -4$, $y = 62$, $z = 16$; FWE $p < 0.05$, Fig. 3b) in this statistical map. A mediation analysis revealed the degree to which VMPFC and RMPFC signals encode accFB value significantly mediated age-related increases in the accFB influence (i.e., indirect effect of VMPFC: $B = 0.02$, SE = 0.01, 95% CI: [0.0064–0.0405], indirect effect of RMPFC: $B = 0.02$, SE = 0.01, 95% CI: [0.0035–0.0442]; Fig. 3d). Significant mediation effect was also observed with the large RMPFC cluster identified by whole-brain analysis ($B = 0.02$, SE = 0.01, 95% CI: [0.0044–0.0452]). These suggest that VMPFC and RMPFC signals encoding accFB parameter are key mediators of the age-related increase in self-protecting behavior based on the accFB.

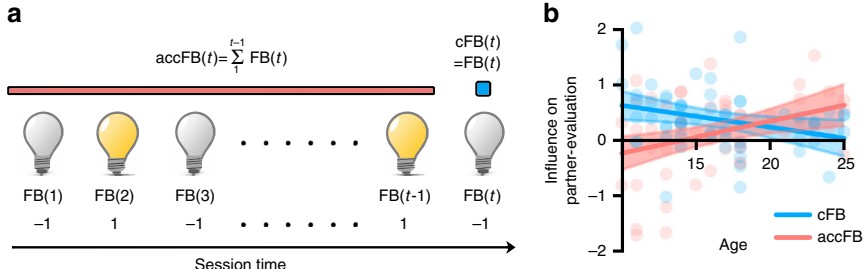

**Fig. 2** Developmental trajectories of the cFB and accFB influence. **a** Schematic illustration of current feedback (cFB) and accumulated feedback (accFB) parameter. Lighted and unlighted bulbs describe the received feedback value of each trial. cFB depicted by blue dot is feedback value of current trial. accFB, depicted by the red line, is accumulated feedback value summed from the first trial to the immediately preceding trial. **b** Interaction between feedback type and age on contribution to partner-evaluation. Blue circles, line and band indicate individuals' cFB parameter estimates, the best fitted line and 95% confidence interval band, respectively. Red circles, line, and band indicate individuals' accFB parameter estimates, best fitted line and 95% confidence interval band, respectively

**Reinforcement learning model**. To further elaborate the neural mechanisms whereby the value of self-protective (VSP) behavior is dynamically updated by social feedback, we fitted a reinforcement learning (RL) model[31] to individuals' partner-evaluation data. This model assumes people would unfavorably evaluate partners when VSP is high (i.e., when received social evaluation (s) turns negative). Three participants who violated this assumption (i.e., those who gave a partner favorable evaluation when their VSP is high) were excluded for this analysis.

In this model (Eq. (2)), VSP is updated by the PE between currently received feedback (i.e., $FB_t$) and previously estimated VSP (i.e., $VSP_{t-1}$) multiplied by learning rate ($\alpha$), a free parameter independently estimated for each individual. According to previous literature[32,33], a high learning rate indicates putting high weight on recent rather than remote feedback and a low learning rate indicates putting uniform weights on all the current and previous feedback (Fig. 4a). Therefore, the learning rate corresponds to the relative use of cFB and accFB, respectively, as confirmed by a post hoc analysis demonstrating that learning rates were associated positively with the cFB influence (Spearman's rank order correlation, $r_s(53) = 0.383$, $p = 0.004$), and negatively with the accFB influence (Spearman's rank order correlation, $r_s(53) = -0.44$, $p = 0.001$).

To confirm self-protective bias in partner-evaluation due to received social feedback regardless of age, we tested whether partner-evaluation bias is primarily driven by partner-derogation with high VSP rather than partner-enhancement with low VSP. The results showed probability of giving positive feedback was significantly lower than the chance level when VSP is high (one-sample Wilcoxon signed rank test, median = 0.35, interquartile range = 0.27; $z = -3.854$, $p < 0.001$), but it was not different from chance level when VSP is low (one-sample Wilcoxon signed rank test, median = 0.51, interquartile range = 0.28; $z = 0.462$, $p = 0.645$) (Supplementary Fig. 3). This indicates that people derogate partners when they received negative feedback, but not necessarily enhance partners when received positive feedback.

Consistent with the developmental trajectories of the cFB and the accFB influence, there was a significant decrease in learning rate with increasing age (Bootstrap linear regression, $B = -0.029$, $p = 0.013$, CI: [−0.05 to −0.006]; Fig. 4b), suggesting a developmental shift, such that younger individuals are more strongly informed by immediate feedback and with age, there is an increasing reliance on cumulative feedback, to choose when to derogate partners. Furthermore, when compared with the median value (0.5) of the entire range of learning rates (0–1), adults from independent larger sample ($N = 28$) showed significantly low learning rate (one-sample Wilcoxon signed rank test, median = 0.1372, interquartile range = 0.62; $z = -2.08$, $p = 0.037$),

suggesting that large amount of feedback integration was observed in the adult population across two independent studies.

To elucidate the neural systems for dynamically updating VSP by social feedback history, individual-specific parametric modulation maps were estimated (GLM2) and regressed against learning rate. We found that RMPFC signals extending into VMPFC (R-VMPFC; BA10/11; 539 voxels at $x = -6$, $y = 62$, $z = 10$; SVC FWE $p < 0.05$) and another VMPFC signals (BA11; 57 voxels at $x = 10$, $y = 50$, $z = -18$; SVC FWE $p < 0.05$) associated inversely with the PE showed negative correlations with individuals' learning rates (Fig. 4c; see Supplementary Fig. 1b for anatomical ROI mask). Exploratory whole-brain analysis identified a region across posterior parietal portex (PPC) (BA23/26/30; 144 voxels at $x = 6$, $y = -54$, $z = 26$; FWE $p < 0.05$, Fig. 4c) in addition to the large cluster of R-VMPFC (BA10/11; 618 voxels at $x = -6$, $y = 62$, $z = 10$; FWE $p < 0.05$). A mediation analysis confirmed that the degree to which R-VMPFC and PPC signals negatively correlated with PE significantly mediated age-related decrease in learning rate (i.e., indirect effect of R-VMPFC (for both SVC and whole-brain corrected clusters): $B = -0.02$, SE = 0.01, 95% CI: [−0.03 to −0.006], indirect effect of PPC: $B = -0.01$, SE = 0.01, 95% CI: [−0.0244 to −0.0031]; Fig. 4d). It should be noted that the mediation effect remained significant even when tested with two separate clusters of RMPFC and VMPFC identified by using heightened initial threshold (i.e., $p = 0.0005$). This indicates that R-VMPFC and PPC signals tracking negative PE are key mediators of the developmental shift in integrating previously received social feedback to calculate the value of self-protective behavior.

## Discussion

The present study used a novel task and a computational-based approach to evaluate the developmental trajectories underlying when social feedback triggers self-protective processes from late childhood to young adulthood. Our results showed that there is a transition in self-protecting behavior such that youth immediately react to momentary evaluation whereas adults' behavior relied on temporally extended cumulative evidence of social evaluation, which was also predicted by RL model. Neuroimaging data revealed that, unlike VMPFC, which is generally involved in self-protecting behavior, RMPFC is recruited for self-protecting behavior based on accumulated feedback history, that is, integrating a large amount of previous feedback for computing the value of self-protection.

Our results showed that feedback history can be a key revealing factor of age-related differences in self-protection against social evaluation. That is, self-protecting behaviors in children are most directly elicited by single social evaluation, whereas self-protective

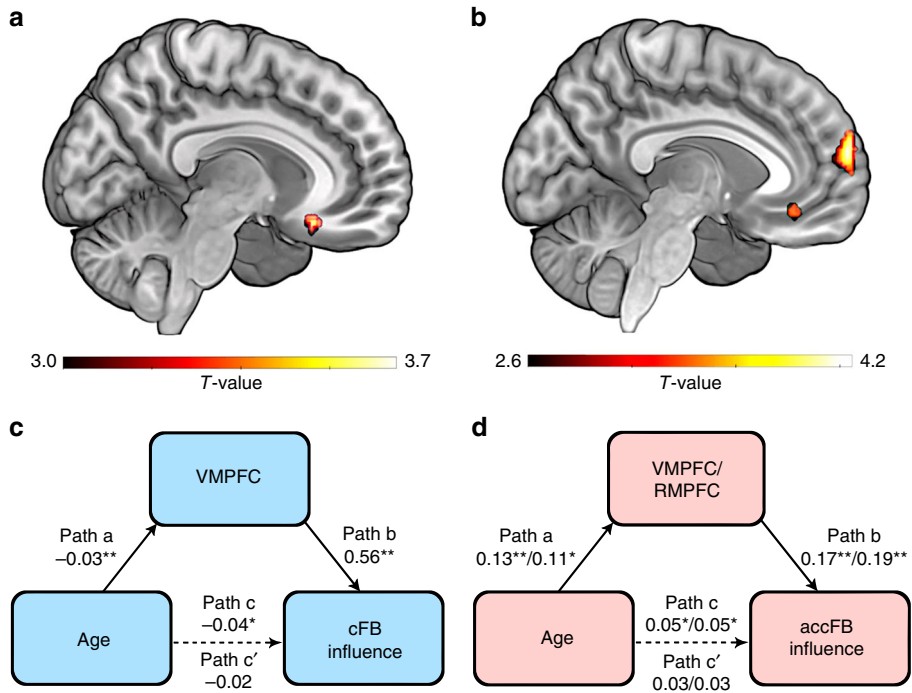

**Fig. 3** Neural mediators of age-related changes in cFB and accFB influence. **a** Statistical parametric map depicting VMPFC (SVC FWE < 0.05) signals differentiating cFB varying from negative to positive values at the feedback receipt event, which correlated positively with individual differences in the cFB influence (see also Supplementary Table 2). **b** Statistical parametric map depicting VMPFC (SVC FWE < 0.05) and RMPFC (whole-brain and SVC FWE < 0.05) signals differentiating accFB varying from negative to positive values at the partner-evaluation event, which correlated positively with individual differences in the accFB influence (see also Supplementary Table 2). **c** Age-related decrease in the cFB influence was mediated by the cFB-related parametric modulation estimates of VMPFC (**a**) (The numbers indicate regression coefficients. $^*p < 0.05$, $^{**}p < 0.01$; Bootstrapping mediation analysis). **d** Age-related increase in the accFB influence was mediated by the accFB-related parametric modulation estimates of VMPFC and RMPFC (**b**) (Regression coefficients before and after the slash correspond to VMPFC and RMPFC, respectively. $^*p < 0.05$, $^{**}p < 0.01$; Bootstrapping mediation analysis)

behaviors in adults are influenced primarily by how they have been evaluated in general by people they have previously encountered. This finding is in line with literature indicating those in late childhood and early adolescents are highly vulnerable to and easily perturbed by social evaluation[7,8,11], whereas adults have more stable self-esteem[13] and are able to regulate self-protective behavior by integrating previous social experiences[5,25].

Why do adults rely more on accumulated rather than immediate social feedback for self-protection? Based on previous work on age-related increases in the sophistication of emotion regulation tactics in adult populations[34–36], it can be speculated that social experiences accumulated during adolescence may contribute to forming more sophisticated self-protective tactics, which can then buffer them from reacting to immediate social-evaluative feedback. Specifically, people may have realized that averaging feedback from multiple others, as opposed to a single instance of feedback from one particular individual, would provide more reliable information for estimating one's social standing. Alternatively, people would learn to internalize or hide aggression in response to social threat because impulsive defensive response to immediate feedback may harm one's reputation[37,38]. In doing so, accumulated positive social feedback in the past can act as a buffer against impulsive aggression toward a person who gave a negative feedback. In addition, a number of nonsocial domains have revealed continuing development of future orientation throughout adolescence, as observed in delay discounting tasks[39,40].

Neuroimaging data revealed that self-protection triggered by both immediate feedback and accumulated feedback was supported by scaled activity in the VMPFC, whose activity following

social feedback has been shown to predict self-protective behavior[4], and individual variability in self-esteem[23]. In general, VMPFC has been implicated in the regulation of stressful as well as rewarding events in both animals[41,42] and humans[43,44], and the guidance of adaptive decisions under situational constraints[45]. In this context, the VMPFC may detect potentially meaningful social-evaluative cues (i.e., current or accumulated feedback) and prompt self-protective behavior to avoid anticipated decrease in self-value. This finding, combined with previous studies, suggests that VMPFC may be a primary brain structure that is engaged by default to compute the value of self-protection in response to social threats from the environment and to trigger adaptive behavior.

Unlike the immediate feedback effect, self-protection based on accumulated feedback recruited RMPFC as well as VMPFC. It is noteworthy that RMPFC was exclusively engaged in the accumulated feedback effect, which requires the integration of temporally remote social information and sophisticated expression of self-protective motivation in an indirect and delayed manner. In line with it, RMPFC has been shown to track cumulative outcomes on a trial-by-trial basis[27], to adaptively adjust behavior based on episodic information currently unavailable[46,47], to compute the value of long-sighted decisions[46,48], to flexibly manage conflicting goals[49,50], and to ruminate on anger-provoking or self-related events[51]. Moreover, supporting the age-related increase in the RMPFC function mediating the accumulated feedback effect, studies have shown protracted maturation of RMPFC[52–55] and its connectivity with other regions[56,57] throughout adolescence. Importantly, such a functional and structural development of RMPFC has been

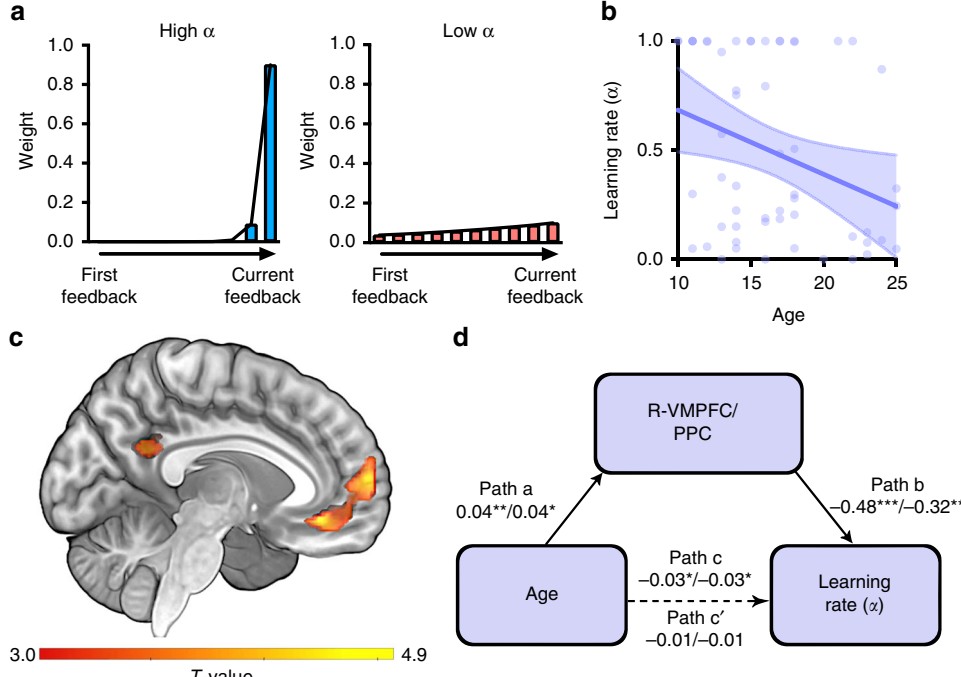

**Fig. 4** Analyses with reinforcement learning (RL) model. **a** Illustration of the relationship between the learning rates (high and low) from RL model and differential impact of the current and previous feedback on partner-evaluation at the current trial (tenth trial in this illustration; for mathematical illustration, see Eq. (3)). **b** Negative association between age and learning rate. Purple circles, line, and band indicate individuals' learning rates, the best fitted line and 95% confidence interval band, respectively. **c** Statistical parametric map depicting RMPFC extending into VMPFC (whole brain and SVC FWE < 0.05) signals and PPC (whole-brain FWE < 0.05) coding prediction error (PE) signals which were negatively associated with learning rate (see also Supplementary Table 2). **d** Age-related decrease in learning rate was mediated by the PE-related parametric modulation estimates of R-VMPFC and PPC (**c**). (Regression coefficients before and after the slash correspond to R-VMPFC and PPC, respectively. *$p$ < 0.05, **$p$ < 0.01; Bootstrapping mediation analysis)

accompanied by increase in abstract thinking such as integration of distal information[28], adaptive coping responses under stress[58], delay of gratification[56], and self-conscious emotion[59].

A mechanistic explanation for the development of accumulated feedback effect was further informed by RL modeling. By focusing on age-related change in learning rate, the present study further extends the previous findings that, in adults, the MPFC encodes the discrepancy between others' and one's own evaluation about oneself to predict the degree of self-enhancing cognitive distortion[26], to compute PE signals when receiving successive social outcomes[25,60], and to dynamically update one's self-esteem based on social feedback[22]. MPFC including both VMPFC and RMPFC, together with PPC, comprises the cortical midline structures (CMS), which has long been recognized as self-referential areas[61,62]. The present study further elucidates the function of this network in deeply encoding and incorporating moment-to-moment social-evaluative cues into preestablished self-value. Specifically, functional maturation of this network to encode dynamic social-feedback prediction error contributed to age-related increases in the extent to which self-relevant social evidence are temporally accumulated (i.e., learning rate). Such a development of the CMS function in dynamic self-construction, potentially accompanied by strengthened interconnectivity within this network[63–65], would provide a more comprehensive and mechanistic explanation of development of sense of self-continuity and stable self-identity[64,66].

The present study provides the behavioral and neural evidence for developmental progress supporting age-related shifts toward more global integration of social feedback triggering self-protection, in which RMPFC plays a critical role. A future study

including nonsocial-evaluative feedback (e.g., from a computer rather than a human) would further elucidate whether the RMPFC is uniquely involved in self-protective behavior due to social feedback. Moreover, given the cross-sectional nature of the present study, a future longitudinal study accompanied by additional collection of hormonal levels, structural brain maturation, and environmental factors, would be needed to have a more complete view of intra-individual developmental changes in self-protective behavior.

## Methods

**Participants**. Participants were recruited through local advertisements and a University community website. Two participants were excluded because of uniform button press or near-uniform button press during the task, which made it impossible to fit their behavioral data to a logistic regression model. A total of 58 participants' behavioral data were analyzed. Supplementary Table 1 shows participants' demographic characteristics including age and gender (mean age = 16.4, range of age: 10–25, and 29 males and 29 females). One participant with excessive and sudden head movement (>1 voxel) was additionally excluded for functional magnetic resonance imaging (fMRI) data analysis. All participants and caregivers of minor participants gave written consent prior to participation in the study, and were compensated with KRW 45,000. The study design and the collection of data complied with all relevant ethical regulations and were approved by Korea University Institutional Review Board.

**Experiment overview**. Participants created artworks in the lab 2 weeks before MRI scanning. At their second visit, they performed a reciprocal artwork evaluation task with 75 supposed partners during fMRI scanning. In each trial, they judged a partner's artwork after their own artwork had been judged by the same partner. This design allowed us to examine how participants' judgments are biased by previously received feedback.

**First visit**. To prevent participants from noticing the experimental purpose, we informed participants that this experiment aimed to investigate how similarity between two people's verbal creativity influences judgment of each other's visual creativity.

Participants were instructed to make a creative artwork and ostensibly told that its creativity would be evaluated by other similar-aged participants who would return to the lab to complete the task earlier than them. Specifically, participants were instructed to make an artwork that was as creative as possible with a set of everyday materials including lettuce, bok choy, beans, chocolate candy, nails, can caps, broccoli, screwdrivers, pliers, jujube, mini peppers, chips, cookies, and tea bags (Fig. 1a). They were also instructed to provide a simple (within five Korean characters) and comprehensive title of the artwork, within the time limit of 30 min. All the participants created artworks as instructed, and some representative examples are shown in Fig. 1b.

Following the artwork creation task, participants answered a short questionnaire regarding the task experience and their own artwork (see Supplementary Note 1). For an additional index of developmental progress other than age, participants younger than twenty completed the pubertal developmental scale[67], which is a self-report questionnaire measuring secondary sex characteristics.

To increase the credibility of the cover story, participants completed the remote associates test[68], a verbal creativity task, at the end of the first visit. In this task, participants were instructed to generate as many words as possible that are related to three words (e.g., Hexagon-Man-White) in each of ten questions.

**Stimuli and experimental conditions**. Although the participants believed that the artworks presented in the task paradigm were made by similar-aged participants, they were actually made by a separate group of adults. Ninety-nine fictitious partners' artworks were rated by independent raters ($N = 16$; 8 male, 8 female, mean age = 25.19) in terms of creativity [1 (not creative at all)–5 (very creative)], and 24 artworks were excluded due to large between-rater variability. The remaining 75 artworks were divided into 5 levels based on their mean creativity rating, which was also used as the parameter of OC. Post hoc analysis (see Supplementary Note 2) confirmed that minors in our experiment distinguished the level of creativity of artworks similarly to the independent adult sample.

**Reciprocal artwork evaluation task**. At the second visit, participants were instructed that other similar-aged participants who had made their second visits earlier than them had already evaluated the creativity of their own artwork. In addition, they would view those same individuals' artworks and evaluate the creativity of them. Before entering MRI scanner, they performed two practice trials.

Participants performed the reciprocal artwork evaluation task inside fMRI scanner. They reciprocally evaluated 75 partners' artworks across 75 trials (i.e., one partner for each trial), which took about 25 min.

Each trial consisted of eight distinctive events (Fig. 1c). Participants viewed a fixation cross (2–4 s) followed by an image of their own artwork (0.5 s), the name of the partner for that trial (1–3 s), and then their FB from the partner (2 s). To prevent any possible impact of inferred gender of partner on creativity evaluation during the task, we made the names of partners as gender-neutral as possible.

At the feedback receipt phase, participants supposedly received positive and negative feedback about their artwork from the named individual, which was presented with a picture of lighted (meaning the partner judged their artwork to be creative) or unlighted (meaning the partner judged their artwork to be not creative) bulb. In addition, a neutral feedback was presented with a comment, "Not yet evaluated" indicating that the partner did not yet come for the second visit, or "No response" indicating that the partner's response was absent in the second visit. Among 75 trials, participants received positive feedback in 25 trials, negative feedback in 25 trials, and neutral feedback in 25 trials. Among 25 neutral feedback, 20 trials said "Not yet evaluated", and 5 trials said "No response". We made this subtle variability in neutral feedback to increase the credibility of the cover story that participants would receive feedback from real human partners during the task. However, two kinds of neutral feedback were treated as the same during analysis.

The pairings between all three feedback conditions (i.e., positive, neutral, and negative feedback) and different artworks were counterbalanced across participants. Each participant viewed 5 artworks in each combination of 3 feedback valences and 5 OC levels, resulting in a total of 75 artworks. Each combination of feedback valence and OC level was presented in a fixed pseudorandom order such that there were no more than two consecutive trials of the same condition. In addition, we used a fixed sequence of the condition trials to ensure that all participants experienced the same sequence of feedback and OC levels.

Following the feedback, the trial continued with a fixation cross (1–3 s), the partner's name again (0.5 s), and a view of the partner's artwork (1–3 s), after which the participant was asked to rate the partner's artwork (<10 s). Participants indicated their decision (i.e., whether the partner's artwork is creative or not) using 2 buttons of a 4-button MR-compatible response grip.

After completing the reciprocal artwork evaluation task, participants answered a short questionnaire asking emotional experiences during the task. This procedure was conducted to indirectly probe suspicion about the purpose of the experiment (see Supplementary Note 3). We confirmed that no participant reported any doubt.

Additionally, we measured several trait variables other than age that could have influenced on the observed feedback effects (see Supplementary Note 4).

**fMRI data acquisition**. fMRI data was acquired using a 3.0T Siemens Magnetom Trio MRI scanner with a 12-channel head matrix coil located at the Korea University Brain Imaging Center. T2*-weighted functional images were obtained using gradient-echo echo-planar pulse sequences (repeat time (TR) = 2000 ms; echo time (TE) = 30 ms; flip angle (FA) = 90°, field of view (FOV) = 240 mm, 80 × 80 matrix; 33 slices; voxel size = 3 mm × 3 mm × 3 mm). FMRI blood oxygen-level dependent (BOLD) activity was measured over one functional run, lasting about 25 min. High-resolution T1-weighted (TR = 1900 ms; TE = 2.52 ms; flip angle = 9°; 256 × 256 matrix; 1 × 1 × 1 mm in-plane resolution) and T2-weighted (TR = 3000 ms; TE = 402 ms; 256 × 256 matrix; 1 × 1 × 1 mm in-plane resolution) structural images were collected as well. The stimuli were presented through an MR-compatible liquid-crystal display monitor mounted on a head coil (refresh rate: 85 Hz; display resolution: 800 × 600 pixels; viewing angle: 30° horizontal, 23° vertical).

**Separate adult study**. To see if the behavioral effect observed in adults (i.e., accFB influence) is replicable in an independent sample, we did the same behavioral analysis to the previously obtained behavioral data of an independent sample with 28 adult participants.

In this study, 29 healthy college students (mean age = 23.1, range: 20–27) were recruited through a university community website. One participant was excluded due to reporting a suspicion about the cover story, leaving 28 participants in the final data analysis. The study protocol was approved by the Korea University Institutional Review Board and all participants gave written consent before the experiment. All participants were compensated with KRW 45,000. This separate adult study was almost identical to the main developmental study except for subtle differences in task structures (see Supplementary Note 5).

**Behavioral data analysis**. Analysis of behavioral data aimed to estimate the degree to which participants' decisions were influenced by the value of feedback they had just received from the individual (i.e., cFB), and the accumulated previous feedback they had received throughout the task as a whole (i.e., accFB), controlling for OC of the partner's artwork. We fitted a logistic regression model (Eq. (1)) to individuals' binary partner-evaluative decision (1: creative, 0: not creative) using the Matlab 2015b fitglm function (Mathworks, Natick, MA).

$$\text{Logit}[P(\text{Decision} = 1)] = \beta_{cFB} x_{cFB} + \beta_{accFB} x_{accFB} + \beta_{OC} x_{OC} + \beta_0 \quad (1)$$

cFB is feedback value of current trial ($t$) and accFB is cumulative feedback value summed from the first trial to the immediately preceding trial ($t - 1$) (see Fig. 2a). accFB of the first trial was set to 0.

$$cFB(t) = FB(t)$$
$$accFB(t) = \sum_{1}^{t-1} FB(t)$$

The cFB was parameterized as −1 (negative feedback), 0 (neutral feedback), and +1 (positive feedback). The range of accFB was rescaled from −3 to +3 to −1 to +1 which is comparable with that of cFB. The OC was determined as stated above (see section Stimuli and experimental conditions) and was rescaled from 1 to 5 to −1 to +1 comparable with other variables. We fit this regression model to each participant's rating to estimate parameters representing the degrees to which their ratings were influenced by cFB and accFB.

To test the age effect on cFB and accFB influence, we implemented regression analyses with the independent variable of linear continuous age and the dependent variables of parameter estimates of cFB and accFB. To test for interactive effects between feedback type (2: cFB, accFB) and age, we ran an analysis of covariance (ANCOVA) analysis with an independent categorical variable of feedback type (2: cFB, accFB) and linear continuous age to predict their ratings (i.e., partner-evaluation). To replicate the accFB influence in adults, we fitted equation (1) to the behavioral data obtained from the separate adult study and extracted the parameter estimates of accFB, which were then entered into a one-sample $t$ test.

For descriptive purposes, proportion of negative evaluation throughout the task, proportion of negative evaluation after negative cFB (−1), neutral cFB (0), positive cFB (+1), negative accFB (<0), neutral accFB (=0), positive accFB (>0) of total sample, younger youth (≤16 years old), and older youth (>16 years old) are presented in Supplementary Table 3. In addition, the associations between cFB/accFB influence and measurements obtained for exploratory purposes are presented in Supplementary Table 4.

Additionally, to address the concern that the cFB influence on partner-evaluation of the younger participants could reflect simple copying of the partners' feedback, we did a supplemental behavioral analysis targeting the participants younger than 16 years (i.e., the median age of the sample). In this analysis, we verified that the younger participants' behavior is self-protective rather than copying behavior, by showing that the proportion of matching feedback was significantly above the chance level following negative, but not

positive, feedback (see Supplementary Fig. 4). Further supporting this argument, we also found qualitative evidence of self-protective and emotionally charged nature of the younger participants' cFB effect after examining their answers to the post-scanning open-ended questions about emotional experiences when receiving negative feedback during the main task (Supplementary Note 6).

**Preprocessing of fMRI data**. fMRI data were preprocessed and analyzed using SPM8 (Wellcome Department of Imaging Neuroscience, London, United Kingdom). Images were realigned to the first volume to correct for head motion and a mean image was created for each participant. We inspected motion estimates and determined that all participants in the final sample showed <10% TRs exceeding 2 mm interslice motion. The realigned images were normalized to the standard Montreal Neurological Institute EPI template, resampled to $2 \times 2 \times 2$ mm voxels, and spatially smoothed using a Gaussian kernel with an 8 mm full width at half maximum (FWHM).

**Definition of ROI**. Based on our prediction of the functional role of VMPFC in cFB influence, we conducted an analysis restricting the search to VMPFC for localizing neural correlates of cFB influence. Here, we created VMPFC mask (Supplementary Fig. 1a) by combining bilateral medial orbital frontal regions and rectus regions from the anatomical automatic labeling (AAL) atlas[69]. For localizing neural correlates of accFB influence and PE signals estimated by the RL, we restricted our search to MPFC that includes both VMPFC and RMPFC (Supplementary Fig. 1b). To create the MPFC mask, we combined bilateral medial orbital frontal regions, rectus regions, and superior medial frontal regions from the AAL atlas[69]. All the anatomically defined masks were created by using WFU_pickatlas[70].

**Statistical thresholding in the fMRI analysis**. To correct for multiple comparisons, we firstly ran small volume correction with the a priori ROI, which was then followed by an exploratory analysis with whole-brain gray matter as the search volume.

The statistical threshold was determined by using the Analysis of Functional NeuroImages (AFNI) software[71]. Intrinsic smoothness was estimated by using the spatial autocorrelation function (ACF) option in AFNI's 3dFWHMx, and the resulting ACF estimates were used to run Monte Carlo simulations via AFNI's 3dClustSim. We note that this method does not employ the assumption of Gaussian-shaped spatial ACF of fMRI data, which is recently pointed out to be incorrect[72]. The desired cluster size for surviving multiple comparison at $a < 0.05$ over the ROI or whole-brain gray matter were obtained with initial uncorrected $p$ value of 0.001 (see the last column of Supplementary Table 2 for desired cluster sizes of each brain map).

**fMRI data statistical analysis**. General linear modeling was used to estimate neural responses to the experimental conditions. We created predictor variables for four periods of the task ((1), (2), (5), and (9) below), convolved with the canonical hemodynamic response function. In addition, parametric modulators were used to incorporate the valence of feedback received (cFB; −1 for negative feedback, 0 for neutral feedback, 1 for positive feedback on that trial), and the average valence of accFB that had been received until previous trial (accFB; −1 to +1: rescaled from the original scale, i.e., −3 to +3). These two parametric modulators were paired with the feedback-receipt phase of trials to measure neural responses to the feedback itself, and also with the partner-evaluation phase of trials to measure how previously received feedback modulates neural responses related to subsequent evaluative judgments about partners' artworks. A third parametric modulator (OC) was additionally included in the evaluation phase to represent that trial's objectively judged creativity. As such, there were nine task regressors of interest in GLM1: (1) Onset times of partner's name, (2) Onset times of feedback receipt, (3) cFB parametric modulator paired with (2), (4) accFB parametric modulator paired with (2), (5) onset times of evaluation of partner's artwork, (6) cFB parametric modulator paired with (5), (7) accFB parametric modulator paired with (5), (8) OC parametric modulator paired with (5), and (9) onset times of decision option presentation. In addition, six motion regressors were included. All the parametric modulators were mean-centered and were not serially orthogonalized to allow for unbiased comparison between overlapping BOLD signals[73].

Based on the behavioral findings in the present study, we focused on the following analyses for the cFB influence and the accFB influence: (1) identifying neural regions associated with the feedback effects and (2) testing the mediating role of the identified neural regions in age-related changes in the feedback effects. All the analyses were conducted with continuous age variable and the significance of the behavioral results with a continuous age variable was unchanged without one participant excluded from fMRI analysis.

To identify neural regions associated with the behavioral cFB influence and the accFB influence on ratings, we ran second-level voxel-wise multiple regression analyses where individuals' behavioral cFB and accFB parametric estimates extracted from Eq. (1) were regressed on individuals' cFB and accFB parametric maps at the feedback onset time and the evaluation onset time. To control for individual differences in the degree to which OC influences on the BOLD signals, individuals' OC parameter estimates were also entered as a regressor of no interest.

The resulting statistical images presented were created using MRIcroGL (http://www.mccauslandcenter.sc.edu/mricrogl/).

To test the mediating role of the identified neural regions in age-related changes in the FB effects, we ran the mediation analyses using bootstrapping approach using 5000 samples implemented in SPSS macros[74]. Before testing mediation, we confirmed that age was statistically related to recruitment of VMPFC encoding the cFB value ($r(54) = −0.344$, $p = 0.009$), VMPFC ($r(54) = 0.367$, $p = 0.005$) and RMPFC ($r(54) = 0.305$, $p = 0.02$ for SVC FWE corrected RMPFC and $r(54) = 0.327$, $p = 0.01$ for whole-brain FWE corrected RMPFC) encoding the accFB value. The mediation model included the independent variable of age, the outcome variable of cFB or accFB behavioral parametric estimates, and the mediating variable of the parametric estimates extracted from the neural regions associated with the cFB and the accFB influence. One extreme outlier of brain activity identified by the Grubb's test[75] was excluded for the mediation analysis.

**Computational modeling**. To further characterize the neural mechanisms underlying the cFB and the accFB influence, a model-based fMRI analysis using a computational RL model was conducted. This model-based analysis was aimed at identifying neural regions associated with updating the value of self-protective partner-evaluation by integrating sequentially presented social feedback in a moment-to-moment fashion.

The RL model for updating VSP was implemented as below (Eq. (2)), and we fitted individuals' behavioral data to the model to estimate the parameters necessary for subsequent model-based fMRI analyses.

$$\text{VSP}_t = \text{VSP}_{t-1} + \alpha \times [\text{FB}_t - \text{VSP}_{t-1}] \tag{2}$$

$$\begin{aligned}
\text{VSP}_t &= (1 - \alpha)^t \text{VSP}_0 + (1 - \alpha)^{t-1} \alpha \text{FB}_1 \\
&\quad + (1 - \alpha)^{t-2} \alpha \text{FB}_2 + \ldots + \alpha \text{FB}_t \\
&= (1 - \alpha)^t \text{VSP}_0 + \sum_{i=1}^{t} (1 - \alpha)^{t-i} \alpha \text{FB}_i
\end{aligned} \tag{3}$$

In each trial, VSP is updated by the PE between currently received feedback (FB) and previously updated VSP multiplied by learning rate ($\alpha$). Importantly, we reverse-coded the FB parameters (FB = 1: negative FB, FB = 0: no FB, FB = −1: positive FB), assuming that a negative feedback increases the VSP. According to the Eq. (3), a modified version of the Eq. (2), this model assumes that VSP is estimated by summing all the feedback values with greater weights on recent feedback, and summed weights of all the feedback bound to 1. We set initial VSP (i.e., VSP (0)) to 0.

To obtain individuals' optimal learning rates, we conducted a maximum log-likelihood estimation procedure. Specifically, subject-specific VSP parameters were entered as a variable in a logistic regression model (Eq. (4)) where VSP (controlling for OC) predicts partner-evaluation decisions ($D = 1$ if the participant evaluated the partner's artwork as creative; $D = 0$ if the participant evaluated the partner's artwork as not creative). We repeated this estimation procedure with learning rates ranging from 0 to 1 (with a step size of 0.0001) to choose the optimal alpha with maximum log-likelihood in Eq. (4) for each individual.

$$P(D) = \frac{1}{1 + e^{-\left(\beta_{\text{VSP}}(\text{VSP}) + \beta_{\text{OC}}(\text{OC}) + \beta_0\right)}} \tag{4}$$

Based on individuals' optimal learning rates, we conducted linear regression analyses with a predictor variable of age and a dependent variable of the learning rates. As the residuals of the dependent variable (i.e., learning rate) showed nonnormal distribution ($W = 0.923$, $p = 0.002$), we conducted a nonparametric bootstrap regression analysis using 5000 samples. The association between learning rate and cFB influence and accFB influence were tested by nonparametric spearman's rank order correlation due to non-normality of residuals of learning rate.

We also ran the same modeling analysis for the separate group of adults only ($N = 28$) for replication. Due to the nonnormality of residuals of learning rate ($W = 0.804$, $p < 0.001$), a nonparametric one-sample Wilcoxon signed rank test was performed.

To further confirm that partner-evaluation bias is mainly driven by self-protective motivation, we examined the degree of partner-derogation bias when VSP is high (i.e., VSP > 0) and the degree of partner-enhancement bias when VSP is low (i.e., VSP < 0), separately. For this, trial-by-trial fluctuation of decision probability was estimated by a logistic model where partner-evaluative decision (1: creative, 0: not creative) was predicted by variables of previously determined subject-specific VSP and OC. In each participant, all the trials were then divided into high and low VSP trials and the degrees of partner-evaluation bias in each type of VSP trials were assessed by comparing the mean decision probabilities against the chance level (i.e., probability = 0.5) by using a one-sample Wilcoxon signed rank test (nonparametric test was used due to non-normality of mean decision probabilities when VSP is high). Here, a significantly higher and lower decision probability compared to the chance level would indicate partner-enhancement bias and partner-derogation bias, respectively. Additionally, we tested the difference in decision probability between the two trial types by using a paired Wilcoxon signed rank test.

**Model-based fMRI analysis**. To identify neural regions associated with the parameters estimated by the RL model fitting, we constructed the GLM2, which contains eight task regressors: (1) onset times of partner's name, (2) subject-specific VSP parameters updated in the immediately preceding trial (i.e., $VSP_{t-1}$) paired with (1), (3) onset times of feedback receipt, (4) subject-specific PE parameters computed from the current trial (i.e., $[FB_t - VSP_{t-1}]$) paired with (3), (5) onset times of evaluation of partner's artwork, (6) subject-specific VSP parameters updated in the current trial (i.e., $VSP_t$) paired with (5), (7) OC of the partner's artwork in the current trial paired with (5), and (8) onset times of decision option presentation. In addition, six motion regressors were included in the GLM2. As in the preceding fMRI analysis, all the parametric modulators were mean-centered and were not serially orthogonalized to allow for unbiased comparison between overlapping BOLD signals[73].

In the second-level group analysis, participant-specific learning rates were regressed on the parametric maps of $VSP_{t-1}$ at the partner's name event, those of the PE at the feedback receipt event, and those of $VSP_t$ at the partner-evaluation event.

Then, we conducted a mediation analysis to examine whether the neural regions associated with the model parameters (i.e., PE) indeed mediate the age-related decrease in learning rate (i.e., age-related increase in feedback integration). Before testing mediation, we confirmed that age was statistically related to recruitment of R-VMPFC ($r(51) = 0.374$, $p = 0.006$ for SVC FWE corrected R-VMPFC and $r(51) = 0.379$, $p = 0.005$ for whole-brain FWE corrected R-VMPFC) and PPC ($r(52) = 0.33$, $p = 0.01$). The mediation model included the independent variable of age, the outcome variable of learning rate, and the mediating variable of mean parametric estimates of the neural regions associated significantly with the model parameters. One extreme outlier of R-VMPFC activity identified by the Grubb's test[75] was excluded for the mediation analysis.

**Code availability**. Task stimuli and codes for behavioral task, behavioral analysis, neuroimaging analysis, and computational modeling are available through the Open Science Framework repository[76].

**Data availability**. Unthresholded maps of Figs. 3a, b and 4c are available at Neurovault. Other data are available on request due to privacy or other restrictions.

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

## Acknowledgments

This work was supported by the National Research Foundation of Korea Grant funded by the Korean Government (NRF-2017M3C7A1041822). The authors would like to thank Euisun Kim, Seongji Cho, Jungsun Yoo, and Minsun Kim for their assistance in data collection and Dr. Wi Hoon Jung for his advice for data analysis.

## Author contributions

L.Y. and H.K. designed the research. L.Y. performed the experiments and L.Y. and H.K. analyzed data. L.Y., L.H.S., and H.K. interpreted findings and wrote the manuscript.

## Additional information

**Competing interests:** The authors declare no competing interests.

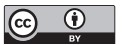

