## [Peer Review File · Nature Communications]

Reviewers' comments:

Reviewer #1 (Remarks to the Author):

In the current study, the authors were interested in how age-related differences in self-protected bias were influenced by short vs. long-term social evaluative feedback and their representation in the brain. Using an artwork evaluation task to elicit social feedback on a trial-by-trial bases, they found that modulation of the vMPFC mediated the relationship between self-defensiveness based on immediate social evaluative feedback. The modulation of rostro-MPFC, on the other hand only mediated the self-defensiveness of accumulated evaluations over many trials. Overall, I liked the approach and paradigm used in the study, and I think the authors are attempting to answer an important question regarding the contribution of different areas of the MPFC in computing self-evaluation based on social feedback and how they may change over time. However, I have some concerns about the strength of the evidence presented here.

My biggest concern with the manuscript is that the all of the fMRI analyses reported in the manuscript were based the use of a small volume correction (SVC) within two 10-mm radius ROIs. SVCs have fallen out of favor among some researchers because they are often seen as a bit dubious. If you have a priori ROIs, why not just run your analyses using those regions? If the exact areas are unknown, why not to a whole-brain analysis? Moreover, although the authors claim that they focused their analyses on these regions, the brain activation patterns displayed in Figures 3 and 4 clearly show activation results well outside of these regions and extending into cerebellum and occipital cortex. The authors should be forthcoming about these analyses if they were indeed conducted. If the SVC results were just a small part of the overall analyses of the fMRI data, I could be a little more forgiving. However, given that all the imaging results are all centered around this analytic procedure, I don't have confidence about what I can take away from this analysis.

Another concern I have is the use of parametric modulator estimate from the fMRI data in the mediation modeling. The PMs can be thought of as an estimate of the slope of the modeled trial-by-trial responses, unlike simple beta maps from standard univariate analysis which measure the magnitude of the response. It seems to me like this would change the interpretation of what the mediation model is telling you. This approach might be completely appropriate; however, it makes interpreting these values within the context of the mediation model a bit confusing to me. I would like to see the authors walk through the logic of these models in more detail.

On a more minor note, the authors should be clear about the cross-sectional nature of their age effects. They discuss "age-related changes", but as in all cross-sectional work, they need to be clear that these are not within-subject changes and caution should be taken when interpreting the age effects.

Reviewer #2 (Remarks to the Author):

In this study the authors have investigated the neural mechanisms related to dealing with negative feedback from others, specifically regarding self-protective responses. For this purpose the authors have developed a new and innovative paradigm (artwork evaluation task) where participants are first asked to create an artwork and then evaluate others' artworks and eventually receive feedback from others on their own artwork. The authors employ computational modeling to examine the brain regions involved in learning from negative feedback. They show that VMPFC is involved in responses in relation to current as well as accumulated feedback whereas RMPFC is particularly involved in responses to accumulated negative feedback history. This is a novel study using a creative task to examine age related differences in responses to negative feedback. However, my main struggle in the manuscript has been with the term 'self-protective behavior' as

I am not convinced that the authors can claim that this is what they are measuring. I will elaborate on this point below.

As much as I like the task that the authors have developed, I do not completely see how the task measures self-protective behaviors. The task is about evaluation of artwork on its creativity. The authors indicate that creativity is a subjective domain that allows room for different opinions. Would this then not make it easy for participants to accept negative feedback or not take it personal? The (negative) feedback refers to the artwork and not to the person.

On a related note, the authors refer to responses to negative 'social' feedback in describing the goal of the study. I assume that the authors see the feedback in the current paradigm as social since it comes from others. However, I see the term 'social feedback' more suitable for, for example, feedback related to acceptance or rejection of an individual, or in social interactions that involve trust or fairness related components. Here the feedback is on the artwork of an individual and they receive feedback from other whom they do not further interact with (so I find it difficult to see the evaluators as 'interaction partners' in this paradigm). In other words, the feedback is not really personal and thus I question to what extent it can be termed as 'social'. Why did the authors choose to have the 'creativity' of the artwork evaluated rather than having others indicate whether they like or dislike the artwork for example? Would this not make the evaluation somewhat more personal and social? In order to claim that the feedback is 'social' the authors should include a non-social condition in the task where for example the computer randomly gives feedback on the artwork.

Regarding the task as a measure of self-protective behavior: the task involves 75 trials of artwork evaluation by 75 different individuals. Here the authors use negative evaluations of artwork that follow negative feedback from the other person as a measure of self-protective behavior. How do the authors distinguish such self-protective behavior from genuine negative judgments of artwork (that is, cases where the participant really finds the artwork not creative, independent of the received negative feedback)? I would think that one can talk about self-protective behavior when the original judgment of an artwork is positive and changes to negative following reception of negative feedback on one's own artwork. In other words, I missed a baseline measurement.

The authors find that with increasing age there is a greater effect of accumulated negative evaluations. But the percentage of received negative feedback was for everyone around 33% (25 out of 75 trials). On p. 8 it is stated 'This indicates that adults engage self-protection by derogating others when accumulated social evaluations turn negative, ...' However, if I understand the task correctly, at no point in the task the accumulated social evaluations are really negative as they are never in majority.

Regarding the age differences that they find in this task: why would younger participants be showing more self-protective behavior to negative feedback? How can the authors know that younger participants are not simply copying other's behavior and repeating the feedback they have received?

Is the vACC region also subgenual ACC? If yes, this region has been implicated in reward processing, also in relation to depressive symptoms. Can the authors discuss the involvement of this region in relation to reward or motivational systems and how this might be involved in the current task?

In their discussion of the findings I missed a discussion of brain development, particularly in the MPFC, in relation to the developmental findings. Several developmental studies suggest decreasing levels of MPFC activity across adolescence (in relation to self-relevance/referral). How do the authors' findings on age related increases in MPFC activity fit with this literature? In general I found the discussion to be lacking a developmental perspective which positions the current findings in the adolescent brain development literature.

Minor points:

Can the authors report on % negative evaluations following the three types of feedback that participants receive and report on links of these with age?

Participants were asked to indicate how creative they think their artwork is as well as their expectation on how creative others would judge their artwork (questions indicated in Supp note 1). Did the authors use these scores in the analyses as a measurement of divergence from one's expectations? I can imagine that the negative feedback might be perceived differently depending on one's expectation of other's reactions on their artwork. It would also be interesting to include descriptive results for the questions described in supp note 1.

Considering that creativity is a subjective domain that allows room for different opinions, why were artworks with large between-rater variability excluded? How were artworks included in the study rated (mean, sd for the OC scores)?

Were there artworks that used the exactly same materials as used by the participant? I can imagine that such similarity in used material might make result in more emotional reactions to negative feedback from the other (e.g., participants might like the artwork more and expect the other person to like their artwork more as well).

Reviewer #3 (Remarks to the Author):

The manuscript reports a study which advances our knowledge of how the MPFC and VACC are implicated in self-protective responses by examining how they relate to "in the moment" threats to self-esteem versus accumulated histories of self-esteem threat. These advances and the inclusion of a (relatively large) sample which goes beyond the usual 18 year old convenience samples are the strengths of this report.

The report would be greatly strengthened, however, from a revision that includes greater precision for certain claims in light of existing literature and theory. For example, the manuscript claims "One question that remains poorly understood is what types of feedback statistics in the environment trigger self-protection: a momentary slight by a peer or a more general sense of contextual rejection accumulated over a longer time horizon? (p 3-4)" This claim is incorrect. Both of these types of threats have been well-characterized in the behavioral literature. The main (and strong) contribution of the current manuscript is to characterize the neural mediation of these two types of threat. However, it is not the first study to show that both types of threat trigger self-protective responses so it would be more accurate to delete that statement.

Similarly, the manuscript states "Despite the evidence alluding to age-related increases in self-protective processes based on social feedback history, its underlying neural mechanism and the developmental trajectory are currently unknown (p. 4)." This is potentially misleading to readers. There has been lots of work in the behavioral literature on how aging affects self-protection (see any work on the Socioemotional Selectivity Theory) and some work (which is cited in the manuscript) characterizing the neural mechanism of self-protective responses in response to "in the moment" threats. The manuscript seems to be claiming the study is the first to break ground in establishing these relations but it is not. Its strength is looking at whether neural regions previously associated with self-protective responses mediate age-related changes suggested by the literature and using computational measures to assess responses to momentary vs accumulated threat. The claim needs to be revised to reflect that known regions are examined in relation to previously suggested age-related changes. Relatedly, the manuscript states that the authors "speculate that people learn to develop more sophisticated ways of protecting themselves

as they age through countless social interactions which buffer them somewhat from moment-to-moment shifts in perceived social inclusion (p. 14)." Again, the entire body of literature on Socioemotional Selectivity Theory as well as work by Blanchard-Fields, Hess, etc. has shown that aging is related to changes of social cognition and selection of social situations which create this buffer. It would be ideal if the manuscript could be revised to incorporate this existing work in relation to the hypotheses and interpretation of the results.

Corrections made through the revision process are highlighted in red both in the main manuscript and the supplementary text. In addition to our answers to the points indicated by the reviewers, we have added minor changes, which are also highlighted in red, and modified the Fig. 4B (i.e., bubble plot without confidence band) to a scatter-plot with confidence band to match with the other scatter plot for main behavioral result (Fig. 2B).

Reviewers' comments:

Reviewer #1 (Remarks to the Author):

In the current study, the authors were interested in how age-related differences in self-protected bias were influenced by short vs. long-term social evaluative feedback and their representation in the brain. Using an artwork evaluation task to elicit social feedback on a trial-by-trial bases, they found that modulation of the vMPFC mediated the relationship between self-defensiveness based on immediate social evaluative feedback. The modulation of rostral-MPFC, on the other hand only mediated the self-defensiveness of accumulated evaluations over many trials. Overall, I liked the approach and paradigm used in the study, and I think the authors are attempting to answer an important question regarding the contribution of different areas of the MPFC in computing self-evaluation based on social feedback and how they may change over time. However, I have some concerns about the strength of the evidence presented here.

My biggest concern with the manuscript is that all of the fMRI analyses reported in the manuscript were based on the use of a small volume correction (SVC) within two 10-mm radius ROIs. SVCs have fallen out of favor among some researchers because they are often seen as a bit dubious. If you have *a priori* ROIs, why not just run your analyses using those regions? If the exact areas are unknown, why not to a whole-brain analysis? Moreover, although the authors claim that they focused their analyses on these regions, the brain activation patterns displayed in Figures 3 and 4 clearly show activation results well outside of these regions and extending into cerebellum and occipital cortex. The authors should be forthcoming about these analyses if they were indeed conducted. If the SVC results were just a small part of the overall analyses of the fMRI data, I could be a little more forgiving. However, given that all the imaging results are all centered around this analytic procedure, I don't have confidence about what I can take away from this analysis.

Thank you for the suggestion and for the opportunity to clarify the analyses that supported our study's conclusions. In fact, there are several analyses that supported our conclusions and in the revised paper, and we have made this clearer in the revised manuscript.

First, we do agree with the reviewer that SVC using a peak voxel coordinate from a previous study is now considered a less robust practice in neuroimaging research, although it is also true that we had an *a priori* hypothesis about the involvement of the RMPFC and VMPFC in self-protective behavior, given the previous relevant literatures. To address the points raised by the reviewer, therefore, we have modified the ROI-based analyses to instead use anatomically defined ROIs (VMPFC for cFB result, MPFC for accFB result and Reinforcement Learning result; presented as Supplementary Figure 1). We detailed how we defined the anatomical ROIs

below as well as in the Methods of the revised manuscript. Please note the change in the results following the application of the anatomical ROIs: In the ROI of the VMPFC, only one cluster remained significant, while two clusters of VMPFC were significant in the original manuscript.

Second, we have run an exploratory whole-brain gray matter search for all the analyses. In these exploratory whole-brain analyses, we found RMPFC for accFB effect (Figure 3c) and RMPFC extending into VMPFC tracking the model prediction error signal (Figure 4c). In the computational modelling analysis (Figure 4c), we additionally found another significant cluster in the PPC (Posterior Parietal Cortex). We reported and interpreted this finding in the abstract, the results and the discussion. In fact, as you can see below, all the key findings, particularly, the engagement of RMPFC in self-protection following accumulated feedback and its functional role in dynamic social feedback integration, survived whole-brain gray matter correction. It was only the VMPFC findings for cFB and accFB effect that survived only the small volume correction. However, VMPFC was our a priori region of interest, because it has been well-known for its role in self-protection/self-enhancement¹⁻⁷ and its sensitivity to social feedback⁸⁻¹⁰.

Third, in the original version of the manuscript, we presented the maps at a more lenient threshold (i.e., $p < 0.005$, uncorrected) only for a display purposes. In the revised manuscript, we now show the statistical maps at the corrected threshold (FWE < 0.05).

Finally, we updated the procedures in the Methods such that the methods now include revised section on multiple comparison correction for fMRI data analysis (i.e., **Method > Statistical thresholding in the fMRI analysis**) and on the anatomical definition of region-of-interest (i.e., **Method > Definition of region-of-interest**).

To summarize, we have made the following changes:

- 1) Adopted a small volume correction using an anatomical region of interest (VMPFC for cFB effect and MPFC for accFB effect and modeling analysis, see Supplementary Figure 1 of revised manuscript).
- 2) Implemented a cluster-wise whole-brain gray matter correction to voxel-wise maps with initial threshold of $p < 0.001$.
- 3) Changed the statistical threshold for visualizing the statistical maps of Figure 3 and 4.
- 4) Updated the procedures in the Methods accordingly.

Another concern I have is the use of parametric modulator estimate from the fMRI data in the mediation modeling. The PMs can be thought of as an estimate of the slope of the modeled trial-by-trial responses, unlike simple beta maps from standard univariate analysis which measure the magnitude of the response. It seems to me like this would change the interpretation of what the mediation model is telling you. This approach might be completely appropriate; however, it makes interpreting these values within the context of the mediation model a bit confusing to me. I would like to see the authors walk through the logic of these models in more detail.

As correctly noted by the reviewer, our parametric modulation analyses for the cFB and the accFB effect and PE Integration estimated the slope of the modelled trial-by-trial responses. Specifically, a low and high slope would mean weak and strong modulation of the activity by the feedback parameters, respectively. For example, individual parameter estimates of cFB effect (Figure 3a, Supplementary Fig.2) indicate the degree to which VMPFC activity encodes the levels of feedback (i.e., negative-control-positive feedback), and individual difference in the slope of such a regression line predicted the behavioral cFB effect (i.e., the degree to which other-evaluation changes as a function of feedback value). Based on the same logic, the mediation effect indicates that age-related decreases in the cFB effect was mediated by the degree to which VMPFC activity strongly encodes the levels of the current feedback (i.e., slope of the regression line). Following the comment from the reviewer, we clarified the interpretation of the mediation results in the **Result** section and **the Figure legend**, as well as the **legend of Supplementary Figure** and **the y-axis of Supplementary Figure 2** as below. We denoted revised parts by making them italics, bold and underlined.

=====

Results section:

Neural mediator of self-protection based on current feedback.

...The resulting statistical maps identified VMPFC (Brodmann Area 11; 23 voxels at $x = 10, y = 24, z = -18$; SVC FWE $p < 0.05$, Fig. 3a and Supplementary Fig. 2a; see Supplementary Fig. 1a for anatomical ROI mask), indicating *the degree to which VMPFC activity negatively correlated with the cFB value (i.e., greater activation to negative cFB value)* at the feedback receipt event predicted individual differences in the cFB influence on subsequent ratings of others. A mediation analysis showed *that the degree to which VMPFC signals strongly encode cFB value* significantly mediated age-related decreases in the cFB influence (Indirect effect: $B = -0.02$, $SE = 0.01$, $95\% CI = [-0.0382, -0.0036]$; Fig. 3b), suggesting that the VMPFC activity *differentiating negative-to-positive cFB* is a key mediator of the age-related decrease in self-protecting behavior based on the current feedback.

Neural mediator of self-protection based on accumulated feedback.

... Results indicated that VMPFC and RMPFC (BA11 and BA10; 23 voxels at $x = -6, y = 36, z = -14$ and 194 voxels at $x = -4, y = 62, z = 16$; SVC FWE $p < 0.05$, Fig. 3c and Supplementary Fig. 2b; see Supplementary Fig. 1b for anatomical ROI mask) *signals negatively correlated with the accFB value (i.e., greater activation to more negative accFB value)* at the other-evaluation event predicted individual differences in the accFB influence... A mediation analysis revealed *the degree to which VMPFC and RMPFC signals strongly encode accFB value* significantly mediated age-related increases in the accFB influence... These suggest that VMPFC and RMPFC signals *encoding* accFB parameter are key mediators of the age-related increase in self-protecting behavior based on the accumulated feedback.

The Figure Legend:

Fig. 3 Neural structures mediating age-related changes in cFB and accFB influence. (a) Resulting statistical parametric map depicting VMPFC (SVC FWE<0.05) signals differentiating negative-to-positive cFB value at the feedback receipt event, which correlated positively with individual differences in the cFB influence (See also Supplementary Table 2). (b) Age-related decrease in the cFB influence was mediated by the cFB-related parametric modulation estimates of VMPFC (Fig. 3a). (The numbers indicate regression coefficients. * $p < 0.05$, ** $p < 0.01$). (c) Resulting statistical parametric map depicting VMPFC (SVC FWE < 0.05) and RMPFC (Whole-brain and SVC FWE < 0.05) signals differentiating negative-to-positive accFB value at other-evaluation event, which correlated positively with individual differences in the accFB influence (See also Supplementary Table 2). (d) Age-related increase in the accFB influence was mediated by the accFB-related parametric modulation estimates of VMPFC and RMPFC (Fig. 3c). (Regression coefficients before the slash correspond to VMPFC and after correspond to RMPFC. * $p < 0.05$, ** $p < 0.01$).

Legend of Supplementary Figure:

Supplementary Figure 2. Relationships between behavioral and neural indices of self-protective biases. (a) Scatter plots showing the relationship between the behavioral cFB indices and the extent to which signals of VMPFC negatively correlated with the cFB value with the influence of accFB and OC controlled for (one outlier in the VMPFC activity was excluded from this plot). (b) Scatter plots showing the relationships between the behavioral accFB indices and the extent to which VMPFC activity (left) and RMPFC activity (right) negatively correlated with the accFB value with the influence cFB and OC controlled for (one outlier in the VMPFC and RMPFC activity was excluded from each plot).

The y-axis of Supplementary Figure 2:

Figure 2a. VMPFC signals differentiating neg-to-pos cFB
Figure 2b. VMPFC signals differentiating neg-to-pos accFB (left), RMPFC signals differentiating neg-to-pos accFB (right).

=====

On a more minor note, the authors should be clear about the cross-sectional nature of their age effects. They discuss “age-related changes”, but as in all cross-sectional work, they need to be clear that these are not within-subject changes and caution should be taken when interpreting the age effects.

Following the reviewer’s comment, we further emphasized this point by adding the following sentences in the last paragraph of Discussion and also by stating clearly that this is a cross-sectional study in the Introduction:

Revised part in Introduction:

“This study uses a cross-sectional design to investigate the underlying neural mediators of such an age-related shift in self-protective behavior.”

Revised part in Discussion:

“Moreover, given that the present study is designed as a cross-sectional one, a future longitudinal study accompanied by additional collection of hormonal levels, structural brain maturation, and environmental factors, would be needed to have a more complete view of intra-individual developmental changes in self-protective behavior.”

Reviewer #2 (Remarks to the Author):

In this study the authors have investigated the neural mechanisms related to dealing with negative feedback from others, specifically regarding self-protective responses. For this purpose the authors have developed a new and innovative paradigm (artwork evaluation task) where participants are first asked to create an artwork and then evaluate others' artworks and eventually receive feedback from others on their own artwork. The authors employ computational modeling to examine the brain regions involved in learning from negative feedback. They show that VMPFC is involved in responses in relation to current as well as accumulated feedback whereas RMPFC is particularly involved in responses to accumulated negative feedback history. This is a novel study using a creative task to examine age related differences in responses to negative feedback. However, my main struggle in the manuscript has been with the term 'self-protective behavior' as I am not convinced that the authors can claim that this is what they are measuring. I will elaborate on this point below.

As much as I like the task that the authors have developed, I do not completely see how the task measures self-protective behaviors. The task is about evaluation of artwork on its creativity. The authors indicate that creativity is a subjective domain that allows room for different opinions. Would this then not make it easy for participants to accept negative feedback or not take it personal? The (negative) feedback refers to the artwork and not to the person.

The reviewer raised a concern that the feedback may not necessarily be an elicitor of self-protection because it targeted artwork, not a person. Although we understand the reviewer's concern, we believe that the feedback should be considered a suitable elicitor of self-protection based on prior research, and based on the data patterns in the present study.

First, because the participant is the creator of artwork under evaluation, the feedback serves as an assessment of the ability, more specifically creativity, of the participant. In fact, in our experimental setting, we intended to simulate the real-world situation where one is assessed

by colleagues, teachers, or bosses in school or workplace for his/her work (not restricted to artwork) and receives feedback about the work from them.

There is a rich tradition in social psychology that delivering evaluative performance-based feedback is a robust way to threaten individuals' self-views and trigger self-protective motivation. For example, the Tier Social Stress Task (TSST)¹¹ induces social stress by offering critical feedback on performance on a verbal and math test. Many studies using TSST¹²⁻¹⁶ revealed that social evaluative situations or social evaluative feedback about one's ability elicits self-protective motivation, self-conscious emotion, and stress-related physiological responses. More generally, there are many other demonstrations supporting the argument that feedback on one's performance elicits self-protective motivation or lowers self-esteem¹⁷⁻²⁰.

Based on these groundings, we hypothesized negative social feedback regarding one's ability (creativity in our task) would be a reliable elicitor of self-protection motivation. Of note, we selected 'creative artwork' as an evaluative domain that could be carried out quickly and repeatedly so as to be feasible within the constraints of event-related fMRI.

In addition, the reviewer raised a concern that receiving negative feedback for one's creativity would be tolerable because judgments about creativity are subjective. We agree with the reviewer that judgment about creativity contains some degree of subjectivity, which we believe was necessary in this study's design to experimentally manipulate the feedback while rendering it believable to participants.

Despite the subjectivity of creativity evaluation, there seems to be good reasons to believe that a negative feedback on creativity (specifically) can induce a threat to self. For example, recent studies have shown that self-reported measures of creativity correlated positively with self-reported self-esteem²¹, and tended to be overestimated compared to objective measures of creativity²². In addition, creative individuals are often perceived as more intelligent than less creative ones²³, and being creative can be thought of as a means to promote one's impression particularly among those with a high impression management tendency²⁴. Based on these findings, therefore, we believe that people have strong motivation to be viewed by others as a creative person, probably because being rated creative can boost one's self-esteem.

Supporting this argument, we collected some open-ended post-test data which indicated that participants reported negative feedback from their partners as a negative emotional experience. Example responses include: "I felt bad because I believe people did not see my artwork in detail.", "I felt bad because I received negative evaluations despite the large effort", "I felt bad as it is usual to feel bad when receiving negative words about one's own work", "I couldn't feel good enough when receiving negative evaluation.", "It was frustrating that some people did not acknowledge my effort", "I felt very bad and irritated", "Although I knew that I didn't do very well, I did not feel good with negative evaluation.", "I felt somewhat bad because some people gave me negative evaluation even though they also did not do well", "I thought some people might not have seen my artwork in detail and gave me negative evaluation, because I drew the artwork very elaborately.", "I felt good because many people gave me good evaluation, but I felt somewhat bad because some people gave me negative evaluation", "I also thought my artwork was not that creative. As it was my artwork, however, I felt somewhat bad

when receiving negative evaluation.” Though qualitative, these findings also increase confidence that the artwork evaluation was a suitable task with which to evoke self-related threat based on evaluative feedback.

On a related note, the authors refer to responses to negative ‘social’ feedback in describing the goal of the study. I assume that the authors see the feedback in the current paradigm as social since it comes from others. However, I see the term ‘social feedback’ more suitable for, for example, feedback related to acceptance or rejection of an individual, or in social interactions that involve trust or fairness related components. Here the feedback is on the artwork of an individual and they receive feedback from other whom they do not further interact with (so I find it difficult to see the evaluators as ‘interaction partners’ in this paradigm). In other words, the feedback is not really personal and thus I question to what extent it can be termed as ‘social’. Why did the authors choose to have the ‘creativity’ of the artwork evaluated rather than having others indicate whether they like or dislike the artwork for example? Would this not make the evaluation somewhat more personal and social? In order to claim that the feedback is ‘social’ the authors should include a non-social condition in the task where for example the computer randomly gives feedback on the artwork.

As the reviewer said (“I assume that the authors see the feedback in the current paradigm as social since it comes from others”), we refer it “social feedback” because the feedback came from other people (specifically similar-aged peers).

The reviewer mentioned that explicit signals of acceptance or rejection by others can be more safely regarded as “social feedback” than performance-related feedback from other people, as used in the present study. This idea is in line with early version of sociometer theory^{25,26}, which suggests that self-esteem is primarily affected by being liked (accepted) or disliked (rejected) by others. However, an updated version of sociometer theory^{27,28} extended the range of feedback that can impact on self-esteem by including respect/admiration by others for being competent and being outperformed/underperformed by others. In addition, the hierometer theory²⁹ suggests both two types of social feedback (i.e., one related to community belongingness and the other related to achievement-related standing) can influence self-esteem. Consistent with these theoretical advances, in the TSST task¹²⁻¹⁶ where participants received negative feedback their performance from judges, participants typically report strong negative emotional experiences and show stress-related hormonal/physiological responses. This theoretical and empirical work jointly suggest that negative evaluative feedback on one’s creativity from other people can be regarded as “*social*” feedback which can elicit self-protection motivation.

We appreciate the reviewer’s suggestion to compare social feedback against non-social feedback (from a computer). Adding such a non-social feedback condition would be useful for isolating psychological/neural components uniquely involved in processing social feedback, although it would also require a more sophisticated cover-story to prevent participants from noticing the purpose of the study. To address this point, we added the following sentence in the

last paragraph of Discussion:

“A future study including non-social evaluative feedback (e.g., from a computer rather than a human) would further elucidate whether the RMPFC is uniquely involved in self-protective behavior due to social feedback.”

The reviewer found the term “interaction” partner rather inappropriate because participants would not actually interact with the partners during the task. We agree with the reviewer, so we changed “interaction partner” into just “partner.”

Participants were asked to evaluate the artworks based on creativity rather than preference (i.e., like or dislike), because we believed that the strong subjectivity of preference judgments would rather undermine self-esteem threats and that creativity judgment would be perceived as more objective than preference judgment. In addition, it was expected that the evaluation of creativity could elicit a greater threat to self-esteem than a simple preference judgment, because participants were explicitly instructed to make their works as creative as possible with the belief that their works will be rated by others later with the same standard of creativity.

Regarding the task as a measure of self-protective behavior: the task involves 75 trials of artwork evaluation by 75 different individuals. Here the authors use negative evaluations of artwork that follow negative feedback from the other person as a measure of self-protective behavior. How do the authors distinguish such self-protective behavior from genuine negative judgments of artwork (that is, cases where the participant really finds the artwork not creative, independent of the received negative feedback)? I would think that one can talk about self-protective behavior when the original judgment of an artwork is positive and changes to negative following reception of negative feedback on one’s own artwork. In other words, I missed a baseline measurement.

As pointed out by the reviewer, it was critical to distinguish bias in judgment due to received feedback value from objective judgement about the artwork. For this goal, our regression model for behavioral analysis (Equation (1)), presented below) included the variable of “OC (Objective Creativity of Artwork)” in addition to cFB and accFB variable. Therefore, the beta estimates of cFB and accFB indicate how much an individual’s decision was biased by current and accumulated feedback, respectively, controlling for the objective creativity of artworks (see also **Method > Behavioral data analysis**).

$$\text{Logit}[P(\text{Decision} = 1)] = \beta_{cFB}x_{cFB} + \beta_{accFB}x_{accFB} + \beta_{OC}x_{OC} + \beta_0 \quad (1)$$

To estimate the Objective Creativity of each individual’s artworks, we recruited 16 independent raters (8 male, 8 female, mean age=25.19) who evaluated the creativity of all the artworks used in the main fMRI study, using a 5-point likert scale (i.e., “how much do you think this artwork creative? [1 (not creative at all) – 5 (very creative)]”) (see also **Method > Stimuli and experimental conditions**). Then, we estimated normative creativity rating of each of the artworks by calculating the mean (and the standard deviation) of creativity ratings collected from the independent raters.

To ensure the reliability of the creativity ratings of the artworks used in the main fMRI study, we adopted the following procedures: First, we excluded 24 artworks with large between-rater variability (i.e., large standard deviation). Second, instead of using the actual mean ratings of individual artworks for OC parameter, we categorized 75 artworks into five levels of creativity (i.e., 15 artworks in each level) and used these levels as OC parameter, to minimize any problem due to potential discrepancy in rating between the two groups.

The authors find that with increasing age there is a greater effect of accumulated negative evaluations. But the percentage of received negative feedback was for everyone around 33% (25 out of 75 trials). On p. 8 it is stated ‘This indicates that adults engage self-protection by derogating others when accumulated social evaluations turn negative, ...’ However, if I understand the task correctly, at no point in the task the accumulated social evaluations are really negative as they are never in majority.

To address the comment from the reviewer, we plotted the time-series of the accFB value across trials below:

The graph above shows the trial-by-trial fluctuation of the accFB parameter, which ranges from -3 to 3 through the task and ends at zero in the end. The sequence of feedback was structured so that participants’ accFB would, at times, be below zero.

Regarding the age differences that they find in this task: why would younger participants be showing more self-protective behavior to negative feedback? How can the authors know that younger participants are not simply copying other’s behavior and repeating the feedback they have received?

We greatly appreciate the reviewer’s comment, which inspired us to investigate further the underlying mechanism of younger participants’ evaluative behavior. Based on the three reasons listed below, we are now confident that the younger participants did not merely copy their partners’ behaviors and that their evaluation clearly reflects emotionally-charged self-protective motivation.

First, we believe that the younger participants’ evaluation matched those of their partners only in the trials where they received negative feedback. As presented in the Supplementary Fig.4 of revised main text, the younger participants derogated their partners when they received a negative feedback, as indicated by the proportion of favorable evaluation significantly lower than the chance level (i.e., 0.5) at the negative cFB ($z = -3.294, p < 0.001$). However, they did not enhance their partners when they received a positive feedback, as indicated by the proportion of favorable evaluation indifferent from the chance level at the positive cFB ($z = 0.765, p = 0.444$). Such a statistical pattern remains the same as we included in the analysis only the group of children from elementary school (≤ 13 years). We added this analysis in the Supplementary Figure 4 and noted it in the behavioral analysis part of the main text as below.

=====
“Additionally, to address the concern that the cFB influence on other-evaluation of the younger participants could reflect simple copying of the partners’ feedback, we did a supplemental behavioral analysis targeting the participants younger than 16 years (i.e., the median age of the sample). In this analysis, we verified that the younger participants’ behavior is self-protective rather than copying behavior, by showing that the proportion of matching feedback was significantly above the chance level following negative, but not positive, feedback (see Supplementary Fig. 4).”
=====

Second, we can ascertain that the younger participants did not merely copy the partners’ behaviors by the highly significant Objective Creativity (OC) effect (i.e., β_{OC} in our logit model across the younger participants was significantly greater than zero ($t(31)=9.155, p<0.001$)). This finding indicates that the main goal of the younger participants during the task was to evaluate the creativity of the partners’ artworks as they had been instructed, rather than merely copying their partners’ behaviors. Such a significant OC effect was also evident when we included in the analysis only the group of children from elementary school (≤ 13 years, $t(16)=6.008, p<0.001$).

Third, the participants’ answers to the open-ended questions during the debriefing provide further support to the argument that the younger participants had been emotionally-charged in response to negative feedback from their partners and their evaluative behaviors had been driven by self-protective motivation: “I felt bad because I believe people did not see my artwork in detail.”, “I felt bad because I received negative evaluations despite the large effort”, “I felt bad as it is usual to feel bad when receiving negative words about one’s own work”, “I couldn’t feel good enough when receiving negative evaluation.”, “It was frustrated that some people did not acknowledge my effort”, “I felt very bad and irritated”, “Although I knew that I didn’t do very well, I did not feel good with negative evaluation.”, “I felt somewhat bad because

some people gave me negative evaluation even though they also did not do well”, “I thought some people might not have seen my artwork in detail and gave me negative evaluation, because I drew the artwork very elaborately.”, “I felt good because many people gave me good evaluation, but I felt somewhat bad because some people gave me negative evaluation”, “I also thought my artwork was not that creative. As it was my artwork, however, I felt somewhat bad when receiving negative evaluation.” We added these answers in the Supplementary Note 6 and denoted in the main text as below:

=====
“Further supporting this argument, we also found qualitative evidence of self-protective and emotionally-charged nature of the younger participants’ cFB effect after examining their answers to the post-task opened-questions about emotional experiences when receiving negative feedback during the main task (Supplementary Note 6).”

=====
Is the vACC region also subgenual ACC? If yes, this region has been implicated in reward processing, also in relation to depressive symptoms. Can the authors discuss the involvement of this region in relation to reward or motivational systems and how this might be involved in the current task?

In the original manuscript, we called the VMPFC cluster in Figure 3a as the vACC, as it is located further posterior to the VMPFC cluster shown in Figure 3c. However, a closer examination of both clusters by overlaying them to Brodmann atlas using MRICron revealed that both VMPFC clusters fall into the Brodmann Area 11. In the revised manuscript, therefore, we decided to label both clusters as VMPFC. In addition, following the suggestion by the reviewer, we added more references and our interpretation of the VMPFC function in terms of its role in the regulation of stressful as well as rewarding events in both in animals and humans as below:

=====
Original Manuscript:

“Neuroimaging data revealed that self-protection triggered by immediate feedback was mainly supported by scaled activity in the VMPFC, whose activity following social feedback has been also shown to predict self-protective behavior⁵ and individual variability in self-esteem²⁶. In general, VMPFC has been implicated³⁸⁻⁴⁰ for guiding adaptive decisions under situational constraints⁴¹. These findings combined together suggest that VMPFC may be a primary brain structure that is engaged by default to compute the value of self-protection by incorporating social information from environment.”

Revised Manuscript:

“Neuroimaging data revealed that self-protection triggered by both immediate feedback and accumulated feedback was supported by scaled activity in the VMPFC, whose activity

following social feedback has been shown to predict self-protective behavior⁵ and individual variability in self-esteem²⁶. In general, VMPFC has been implicated in the regulation of stressful as well as rewarding events in both animals⁴⁹⁻⁵¹ and humans⁵²⁻⁵⁵ and the guidance of adaptive decisions under situational constraints⁵⁶. In this context, the VMPFC may detect potentially meaningful social evaluative cues (i.e., current or accumulated feedback) and prompt self-protective behavior to avoid anticipated decrease in self-value. This finding, combined with previous studies, suggests that VMPFC may be a primary brain structure that is engaged by default to compute the value of self-protection in response to social threats from the environment and to trigger adaptive behavior.”

=====

In their discussion of the findings I missed a discussion of brain development, particularly in the MPFC, in relation to the developmental findings. Several developmental studies suggest decreasing levels of MPFC activity across adolescence (in relation to self-relevance/referral). How do the authors' findings on age related increases in MPFC activity fit with this literature? In general I found the discussion to be lacking a developmental perspective which positions the current findings in the adolescent brain development literature.

As pointed out by the reviewer, it is important to explain how we can resolve our age-related increase in RMPFC activity in accFB effect with the several previous the developmental aspects of the same region. In fact, several studies have indicated age-related decrease in RMPFC activity during self-reflection^{30,31}, which is seemingly in conflict with the present study. To reconcile this conflict, we suggest different mental functions required for accumulated feedback effect and retrieving self-knowledge. More specifically, retrieving self-knowledge may require a relatively simpler mental process than that required by the accumulated feedback effect because the latter necessitates accurately tracking sequentially presented episodic information and sophisticated expression of self-protection in an indirect manner. Therefore, the RMPFC would be only weakly engaged during self-referential task in adults because such a processing requires less cognitive effort in adults compared to adolescents. On the other hand, the same region would be more actively engaged in the expression of the accumulated feedback effect in adults compared to adolescents, because the task requires a more sophisticated, complex, and recently acquired mental function. In summary, an age-related change in the RMPFC function would vary depending on the degree of sophistication and complexity in the expression of social behavior.

Following the reviewer's suggestions, we revised and added the paragraph below to discuss our results within the scope of developmental cognitive neuroscience. In addition, as mentioned above, we changed the discussion about the results from the analysis using the reinforcement learning model to address the cluster in the posterior parietal cortex found after adopting whole-brain cluster-wise correction.

=====

Original manuscript:

“Self-protection due to accumulated feedback recruited VMPFC as well as RMPFC. It is noteworthy that RMPFC was exclusively engaged for the accumulated feedback effect, which requires the integration of temporally distributed relevant social information. Consistent with such a role for integrating temporally distributed information, RMPFC has been shown to encode cumulative outcome on a trial-by-trial basis²⁹, to adaptively adjust behavior based on integrated representation of episodic information currently unavailable, and to compute the value of long-sighted decisions⁴²⁻⁴⁴. In addition, the frontopolar cortex (FPC), located adjacent to RMPFC, maintains representations of past experiences that can be utilized for future goals⁴⁵⁻⁵⁰. The present study suggests that such a role of RMPFC in temporal integration of information can be extended to social domain as it matures through development³⁰.

It should be noted that the accumulated feedback effect was associated with both RMPFC and VMPFC, which may indicate that such an effect requires intimate interactions between the two MPFC subregions. Our working hypothesis is that VMPFC intuitively computes the value of self-protection upon receiving negative social feedback, whereas RMPFC is engaged to regulate/override the VMPFC function whenever it detects a conflict/discrepancy between such an intuitive value of self-protection and other higher cognitive goals, which then leads to a more tactical and unnoticeable self-protective behavior. This hypothesis can be further supported by the model-based fMRI analysis showing that RMPFC, along with VMPFC, computes the prediction error between currently received feedback and previously estimated integrated social feedback, which then contributes to computing the value of self-protection. In line with this hypothesis, RMPFC and/or VMPFC has been also shown to encode the discrepancy between others’ and one’s own evaluation about oneself to predict the degree of self-enhancing cognitive distortion, that is, optimistic bias²⁸, to compute prediction error signals when receiving successive social outcomes^{19,20}, and to dynamically update one’s self-esteem based on social feedback²¹.”

Revised manuscript:

“Unlike the immediate feedback effect, self-protection based on accumulated feedback recruited RMPFC as well as VMPFC. It is noteworthy that RMPFC was exclusively engaged for the accumulated feedback effect, which can be characterized by the following features. First, it requires the integration of temporally remote social information. Second, it necessitates a sophisticated expression of self-protective motivation in an indirect and delayed manner^{46,57,58}. In line with it, RMPFC has been shown to track cumulative outcomes on a trial-by-trial basis³⁰, to adaptively adjust behavior based on episodic information currently unavailable^{59,60}, to compute the value of long-sighted decisions⁶⁰⁻⁶², to flexibly manage conflicting goals^{63,64}, and to ruminate on anger-provoking or self-related events⁶⁵. Moreover, supporting the age-related increase in the RMPFC function mediating the accumulated feedback effect, studies have shown protracted maturation of RMPFC⁶⁶⁻⁷⁰ and its connectivity with other regions^{71,72} throughout adolescence. Importantly, such a functional and structural development of RMPFC has been accompanied by increase in abstract thinking such as integration of distal information³¹, adaptive coping responses under stress⁷³, delay of gratification⁷¹, and self-conscious emotion⁷⁴.

A mechanistic explanation for the development of accumulated feedback effect was further informed by reinforcement learning modelling. Results indicated that ventral and rostral medial prefrontal cortex as well as posterior parietal cortex (PPC) which comprise cortical midline structures (CMS), played a significant role. Specifically, functional maturation of this network to encode dynamic social prediction error contributed to age-related increases in the extent to which self-relevant social evidence are temporally accumulated (i.e., learning rate). By focusing on age-related change in learning rate, the present study further extends the previous findings that, in adults, the cortical midline structure, particularly MPFC, encode the discrepancy between others' and one's own evaluation about oneself to predict the degree of self-enhancing cognitive distortion²⁹, to compute prediction error signals when receiving successive social outcomes^{28,75}, and to dynamically update one's self-esteem based on social feedback²⁴. As CMS have long been recognized as self-referential areas^{76,77}, the present study further elucidates the function of this network in deeply encoding and incorporating moment-to-moment social evaluative cues into pre-established self-value. Such a development of CMS function in dynamic self-construction, potentially accompanied by strengthened interconnectivity within this network⁷⁸⁻⁸¹, would provide a more comprehensive and mechanistic explanation of development of sense of self-continuity and stable self-identity^{79,82,83}."

Minor points:

Can the authors report on % negative evaluations following the three types of feedback that participants receive and report on links of these with age?

We reported the proportion of 'positive' evaluation following each of feedback value in the Supplementary Table 3 of our original submission. Following the suggestion by the reviewer, however, we converted the % positive evaluation to % negative evaluation scores. In addition, to address the reviewer's request on linking these to age, we also presented % negative evaluation scores following each feedback value separately for younger participants (age ≤ 16) and older participants (age > 16). It should be noted, however, that this table is only for descriptive purposes and that the exact link between age and feedback influence, after controlling OC, is the main result of the present study, as presented in Figure 2.

Participants were asked to indicate how creative they think their artwork is as well as their expectation on how creative others would judge their artwork (questions indicated in Supp note 1). Did the authors use these scores in the analyses as a measurement of divergence from one's expectations? I can imagine that the negative feedback might be perceived differently depending on one's expectation of other's reactions on their artwork. It would also be interesting to include descriptive results for the questions described in supp note 1.

To address the reviewer’s comment, we added the **Supplementary Table 4** which lists the associations between the behavioral data from the main task and all the other behavioral measures obtained for exploratory purposes, as shown below:

Supplementary Table 4. The relationships between the behavioral indices from the main task (i.e., cFB and accFB effect) and all the other behavioral measure obtained for exploratory purposes. The significance level was Bonferroni-corrected for multiple tests of correlation and no significant association was found with any behavioral measure (the values indicate correlation coefficients). For specific contents of the subjective reports on task experiences and task-related personality trait, see Supplementary Note 1 and Note 4.

Exploratory measurements		cFB	accFB
Additional index for developmental progress	Pubertal Developmental Scale (PDS)	-0.229	0.237
Task experience	Interest	0.014	-0.206
	Effort	0.156	-0.179
	Self-Evaluation	0.132	-0.162
	Expectation of Other-Evaluation	0.044	0.029
Task-related personality trait variables	Perceived significance of creativity	0.143	-0.087
	Need for Approval	-0.105	0.021
	Self-Esteem	0.139	-0.355

Moreover, following the suggestion from the reviewer, we examined the impact of individual differences in “self-evaluation” and “expectation of other-evaluation,” in a new multiple regression analysis with dependent variables of cFB effect or accFB effect and predictor variables of age, self-evaluation, and expectation of other-evaluation. The results showed that for both cFB and accFB effect, age remained the only significant predictor even after controlling for the other individual differences variables.

Considering that creativity is a subjective domain that allows room for different opinions, why were artworks with large between-rater variability excluded? How were artworks included in the study rated (mean, sd for the OC scores)?

As stated in the answer to the third question from the reviewer #2, to estimate the objective creativity of the artworks (i.e., OC parameter), we recruited independent raters ($N = 16$; 8 male, 8 female, mean age = 25.19) who rated the creativity of 99 artworks prior to the main experiment (see also **Method > Stimuli and experimental conditions**). We asked, “how creative do you think this artwork is? [1 (not creative at all) – 5 (very creative)]” Then, we

calculated the mean and the standard deviation of creativity ratings of each of 99 artworks across the 16 independent raters. Since we use OC ratings from independent raters as a proxy for what the same ratings of participants in the main experiment would have been if they had been asked, we wanted these ratings to be as high-confidence as possible. For this, we excluded 24 artworks with large between-rater variability (i.e., large standard deviation) to increase the reliability of the OC parameter.

Were there artworks that used the exactly same materials as used by the participant? I can imagine that such similarity in used material might make result in more emotional reactions to negative feedback from the other (e.g., participants might like the artwork more and expect the other person to like their artwork more as well).

Following the reviewer's comment, we examined the impact on the decisions of the feedback from those who used the same materials as used by the participant. Among 58 participants, 24 participants did not encounter any artwork with the same material as theirs, 18 participants saw one, 11 participants saw two, and 5 participants saw three such artworks. This means that the participants encountered less than 1 artwork on average that is made with the same materials as used for his/her own artwork. Therefore, we believe that the impact on the decisions of the feedback from those who used the same materials as used by the participant is negligible.

However, the reviewer raised quite an interesting and potentially important question, which we believe deserves future studies that are specifically designed to address the question.

Reviewer #3 (Remarks to the Author):

The manuscript reports a study which advances our knowledge of how the MPFC and VACC are implicated in self-protective responses by examining how they relate to "in the moment" threats to self-esteem versus accumulated histories of self-esteem threat. These advances and the inclusion of a (relatively large) sample which goes beyond the usual 18 year old convenience samples are the strengths of this report.

The report would be greatly strengthened, however, from a revision that includes greater precision for certain claims in light of existing literature and theory. For example, the manuscript claims "One question that remains poorly understood is what types of feedback statistics in the environment" trigger" self-protection: a momentary slight by a peer or a more general sense of contextual rejection accumulated over a longer time horizon? (p 3-4)" This claim is incorrect. Both of these types of threats have been well-characterized in the behavioral literature. The main (and strong) contribution of the current manuscript is to characterize the neural mediation of these two types of threat. However, it is not the first study to show that both types of threat trigger self-protective responses so it would be more accurate to delete that statement.

As the reviewer pointed out, both behavioral effects (i.e., self-protective reaction after an immediate feedback and self-protective reaction after accumulated feedback) had been

previously investigated. Accordingly, we decided not to claim strongly about the novelty of the self-protective behavioral effects observed in the present study. However, we believe that the accumulated feedback effect is less well-documented than the immediate feedback effect, and that simultaneous measurement of both effects have been rarely done in a repeated-measures design. Thus, we modified the introduction and emphasized this point as below:

Original Manuscript:

“While self-protective phenomena have been established in a variety of laboratory settings, critical questions remain regarding the nature and mechanisms of self-protective biases. One question that remains poorly understood is what types of feedback statistics in the environment “trigger” self-protection: a momentary slight by a peer, or a more general sense of contextual rejection accumulated over a longer time horizon? The present study deconfounds momentary and accumulated feedback to evaluate its influence on other-derogation.”

Revised Manuscript:

“Although self-protective behaviors have been extensively investigated in a wide range of laboratory-based studies^{2,5}, many of the studies have focused on immediate response to a single instance of social feedback, while a few studies⁶ have characterized self-protection triggered by general sense of being rejected/accepted accumulated over a long time horizon. In addition, simultaneous measurement of immediate and accumulated feedback effects has rarely been done in a repeated-measures design. The present study is designed to examine relative impact of immediate and accumulated feedback on other evaluation, defined as self-protective behavior, in a single participant, focusing on age-related differences in such behaviors. We predicted that, comparing individuals from late childhood to early adulthood, younger individuals would utilize immediate feedback to trigger self-protection, whereas increasing age would predict a shift in reliance on accumulated, rather than immediate, feedback.”

Similarly, the manuscript states "Despite the evidence alluding to age-related increases in self-protective processes based on social feedback history, its underlying neural mechanism and the developmental trajectory are currently unknown (p. 4)." This is potentially misleading to readers. There has been lots of work in the behavioral literature on how aging affects self-protection (see any work on the Socioemotional Selectivity Theory) and some work (which is cited in the manuscript) characterizing the neural mechanism of self-protective responses in response to "in the moment" threats. The manuscript seems to be claiming the study is the first to break ground in establishing these relations but it is not. Its strength is looking at whether neural regions previously associated with self-protective responses mediate age-related changes suggested by the literature and using computational measures to assess responses to momentary vs accumulated threat. The claim needs to be revised to reflect that known regions are examined in related to previously suggested age-related changes. Relatedly, the manuscript states that the

authors "speculate that people learn to develop more sophisticated ways of protecting themselves as they age through countless social interactions which buffer them somewhat from moment-to-moment shifts in perceived social inclusion (p. 14)." Again, the entire body of literature on Socioemotional Selectivity Theory as well as work by Blanchard-Fields, Hess, etc. has shown that aging is related to changes of social cognition and selection of social situations which create this buffer. It would be ideal if the manuscript could be revised to incorporate this existing work in relation to the hypotheses and interpretation of the results.

Following the reviewer's comment, we have reviewed the literature on "Socioemotional Selectivity Theory (SST)." As mentioned by the reviewer, the SST contends aging-related shifts in emotion-regulation strategies. Specifically, older adults (vs. younger adults), who perceive remaining time as limited, tend to construct/select environments that can foster positive affect and to use passive emotion-regulation strategy (e.g., avoidance-denial-escape, suppression of emotions) rather than proactive emotion-regulation (e.g., confrontation) in dealing with interpersonal conflict. Despite all the merits of SST in formalizing age-related changes in coping behavior, most of empirical evidence supporting this theory has been obtained from the samples of older and young adults³²⁻³⁷. Even in some studies with adolescents included³⁸⁻⁴⁰, significant differences were found between older and younger adults but not between young adults and adolescents, which are main age-groups investigated in our study. Therefore, although SST is a theory of life-span social motivation change, its main focus seems to be on the dramatic changes from young adults to older adults rather than late childhood to young adults⁴¹.

To address the concerns raised by the reviewer, we cited the previous literatures on the SST theory in the Discussion. In addition, we added more relevant studies with the age groups similar to that of our study (i.e., childhood to young adulthood), which reported age-related changes in strategic knowledge and aggression tactics in the Introduction. Finally, we revised the manuscript to emphasize the scope of our study is limited to the early part of the lifespan ranging from late childhood to early adulthood, rather than age-related changes across the entire lifespan throughout the manuscript.

=====

<Discussion>

Original manuscript:

"Why do adults rely more on accumulated rather than immediate social feedback for self-protection? We speculate that people learn to develop more sophisticated ways of protecting themselves as they age through countless social interactions, which buffer them somewhat from moment-to-moment shifts in perceived social inclusion."

Revised manuscript:

"Why do adults rely more on accumulated rather than immediate social feedback for self-protection? Based on the previous work on age-related increases in the sophistication of

emotion-regulation tactics in adult populations³⁹⁻⁴⁵, it can be speculated that social experiences accumulated during adolescence may contribute to forming more sophisticated self-protective tactics, which can then buffer them from reacting to immediate social evaluative feedback.”

=====

=====

<Introduction>

Original manuscript:

“... Further, we predict that younger individuals will utilize a shorter span of feedback history to trigger self-protection. Supporting this hypothesis, it was shown that those in late childhood and early adolescents display highly variable state self-esteem responding to immediate context^{12,13}, whereas adults show more stable state self-esteem¹⁴ and build self-concepts based on abundant past experiences¹⁵⁻¹⁸. In addition, recent studies also demonstrated that adults utilize social outcome history to compute social value of self¹⁹⁻²² and adults’ aggression toward an innocent other was found to be modulated by the number of people who previously accepted/rejected her/him⁶.”

Revised manuscript:

“... We predicted that, comparing individuals from late childhood to early adulthood, younger individuals would utilize immediate feedback to trigger self-protection, whereas increasing age would predict a shift in reliance on accumulated, rather than immediate, feedback. Supporting this hypothesis, it was shown that those in late childhood and early adolescents display highly variable state self-esteem responding to immediate context^{12,13}, whereas adults show more stable state self-esteem¹⁴ and build self-concepts based on abundant past experiences¹⁵⁻¹⁸. Moreover, other studies have reported that, throughout adolescence, people develop more sophisticated knowledge of strategies for dealing with interpersonal conflicts¹⁹ and shift from simple/direct to strategic/indirect tactics for expressing aggression^{6,20,21}.”

=====

Also, as suggested by the reviewer, we revised the following paragraph to avoid misleading readers:

=====

Original manuscript:

“Despite the evidence alluding to age-related increases in self-protective processes based on social feedback history, its underlying neural mechanism and the developmental trajectory are

currently unknown. In this study, therefore, we aimed to investigate the neural mechanisms mediating the developmental trajectory of self-protective behavior as measured by biases in other-evaluation. We had an a priori prediction that the medial prefrontal cortex.....”

Revised manuscript:

“This study uses a cross-sectional design to investigate the underlying neural mediators of such an age-related shift in self-protective behavior.”

We apologize for failing to communicate the novelty of the present study more clearly. As correctly pointed out by the reviewer, the behavioral as well as the neural evidence for the immediate feedback effect have been relatively well established in the previous literatures. In addition, the behavioral evidence for the accumulated effect in adults has been also reported before (e.g., Dewall et al., 2010). However, the age-related increase in the accumulated feedback effect has been only indirectly reported before, and our study focused on and confirmed this effect in a systematic way. The key strength that distinguishes the present study from previous literatures is that we found neural evidence for the accumulated feedback effect and the parallel evidence for behavioral and neural mechanisms underlying age-related shift from immediate to accumulated feedback effect. Given the importance of this point, we now emphasize this novel aspect of the present study more in detail both in the Introduction and the Discussion of the revised manuscript.

- 1 Beer, J. S. This time with motivation: The implications of social neuroscience for research on motivated self- and other-perception (and vice versa). *Motivation and Emotion* **36**, 38-45, doi:10.1007/s11031-011-9259-0 (2012).
- 2 Beer, J. S., Chester, D. S. & Hughes, B. L. Social threat and cognitive load magnify self-enhancement and attenuate self-deprecation. *Journal of Experimental Social Psychology* **49**, 706-711, doi:10.1016/j.jesp.2013.02.017 (2013).
- 3 Delgado, M. R. *et al.* Viewpoints: Dialogues on the functional role of the ventromedial prefrontal cortex. *Nat Neurosci* **19**, 1545-1552, doi:10.1038/nn.4438 (2016).
- 4 Hughes, B. L. & Beer, J. S. Medial orbitofrontal cortex is associated with shifting decision thresholds in self-serving cognition. *Neuroimage* **61**, 889-898, doi:10.1016/j.neuroimage.2012.03.011 (2012).

- 5 Hughes, B. L. & Beer, J. S. Protecting the Self: The Effect of Social-evaluative Threat on Neural Representations of Self. *Journal of Cognitive Neuroscience* **25**, 613-622 (2013).
- 6 Hughes, B. L. & Beer, J. S. Orbitofrontal cortex and anterior cingulate cortex are modulated by motivated social cognition. *Cerebral Cortex* **22**, 1372-1381 (2012).
- 7 Rigney, A. E., Koski, J. E. & Beer, J. S. The functional role of ventral anterior cingulate cortex in social evaluation: disentangling valence from subjectively rewarding opportunities. *Social cognitive and affective neuroscience* **13**, 14-21 (2017).
- 8 Somerville, L. H., Heatherton, T. F. & Kelley, W. M. Anterior cingulate cortex responds differentially to expectancy violation and social rejection. *Nature Neuroscience* **9**, 1007-1008, doi:10.1038/nn1728 (2006).
- 9 Somerville, L. H., Kelley, W. M. & Heatherton, T. F. Self-esteem Modulates Medial Prefrontal Cortical Responses to Evaluative Social Feedback. *Cerebral Cortex* **20**, 3005-3013, doi:10.1093/cercor/bhq049 (2010).
- 10 Heatherton, T. F. Neuroscience of Self and Self-Regulation. *Annual Review of Psychology, Vol 62* **62**, 363-390, doi:10.1146/annurev.psych.121208.131616 (2011).
- 11 Kirschbaum, C., Pirke, K.-M. & Hellhammer, D. H. The 'Trier Social Stress Test'—a tool for investigating psychobiological stress responses in a laboratory setting. *Neuropsychobiology* **28**, 76-81 (1993).
- 12 Dickerson, S. S., Gruenewald, T. L. & Kemeny, M. E. Psychobiological Responses to Social Self Threat: Functional or Detrimental? *Self and Identity* **8**, 270-285, doi:10.1080/15298860802505186 (2009).
- 13 Tang, D. & Schmeichel, B. J. Self-affirmation facilitates cardiovascular recovery following interpersonal evaluation. *Biol Psychol* **104**, 108-115, doi:10.1016/j.biopsycho.2014.11.011 (2015).
- 14 van den Bos, E., Rooij, M., Miers, A. C., Bokhorst, C. L. & Westenberg, P. M. Adolescents' increasing stress response to social evaluation: Pubertal effects on cortisol and alpha-amylase during public speaking. *Child Development* **85**,

- 220-236 (2014).
- 15 Dickerson, S. S. & Kemeny, M. E. Acute stressors and cortisol responses: a theoretical integration and synthesis of laboratory research. *Psychological bulletin* **130**, 355 (2004).
 - 16 Gruenewald, T. L., Kemeny, M. E., Aziz, N. & Fahey, J. L. Acute threat to the social self: Shame, social self-esteem, and cortisol activity. *Psychosomatic Medicine* **66**, 915-924, doi:10.1097/01.psy.0000143639.61693.ef (2004).
 - 17 Campbell, W. K. & Sedikides, C. Self-threat magnifies the self-serving bias: A meta-analytic integration. *Review of general Psychology* **3**, 23 (1999).
 - 18 Campbell, W. K., Baumeister, R. F., Dhavale, D. & Tice, D. M. Responding to major threats to self-esteem: A preliminary, narrative study of ego-shock. *Journal of Social and Clinical Psychology* **22**, 79-96 (2003).
 - 19 vanDellen, M. R., Campbell, W. K., Hoyle, R. H. & Bradfield, E. K. Compensating, Resisting, and Breaking: A Meta-Analytic Examination of Reactions to Self-Esteem Threat. *Personality and Social Psychology Review* **15**, 51-74, doi:10.1177/1088868310372950 (2011).
 - 20 Dedovic, K. *et al.* Psychological, endocrine and neural responses to social evaluation in subclinical depression. *Social cognitive and affective neuroscience* **9**, 1632-1644 (2013).
 - 21 Goldsmith, R. E. & Matherly, T. A. Creativity and self-esteem: A multiple operationalization validity study. *The Journal of psychology* **122**, 47-56 (1988).
 - 22 Park, N. K., Chun, M. Y. & Lee, J. Revisiting individual creativity assessment: Triangulation in subjective and objective assessment methods. *Creativity Research Journal* **28**, 1-10 (2016).
 - 23 MacKinnon, D. Illustrative material for some reflections on the current status of personality assessment with special references to the assessment of creative persons. *graduate students, Department of Psychology, University of Utah, Salt Lake City* (1966).
 - 24 Uziel, L. Look at me, I'm happy and creative: The effect of impression management on behavior in social presence. *Personality and Social Psychology Bulletin* **36**, 1591-1602 (2010).

- 25 Leary, M. R. & Downs, D. L. in *Efficacy, agency, and self-esteem* 123-144 (Springer, 1995).
- 26 Leary, M. R., Terdal, S. K., Tambor, E. S. & Downs, D. L. Self-Esteem as an Interpersonal Monitor - the Sociometer Hypothesis. *Journal of Personality and Social Psychology* **68**, 518-530, doi:Doi 10.1037//0022-3514.68.3.518 (1995).
- 27 Leary, M. R. & Baumeister, R. F. in *Advances in experimental social psychology* Vol. 32 1-62 (Elsevier, 2000).
- 28 Leary, M. R., Cottrell, C. A. & Phillips, M. Deconfounding the effects of dominance and social acceptance on self-esteem. *Journal of Personality and Social Psychology* **81**, 898-909, doi:10.1037//0022-3514.81.5.898 (2001).
- 29 Mahadevan, N., Gregg, A. P., Sedikides, C. & de Waal-Andrews, W. G. Winners, Losers, Insiders, and Outsiders: Comparing Hierometer and Sociometer Theories of Self-Regard. *Front Psychol* **7**, 334, doi:10.3389/fpsyg.2016.00334 (2016).
- 30 Pfeifer, J. H., Lieberman, M. D. & Dapretto, M. "I know you are but what am I?": neural bases of self-and social knowledge retrieval in children and adults. *Journal of Cognitive Neuroscience* **19**, 1323-1337 (2007).
- 31 Pfeifer, J. H. *et al.* Neural correlates of direct and reflected self-appraisals in adolescents and adults: When social perspective-taking informs self-perception. *Child development* **80**, 1016-1038 (2009).
- 32 Scheibe, S. & Carstensen, L. L. Emotional aging: Recent findings and future trends. *The Journals of Gerontology: Series B* **65**, 135-144 (2010).
- 33 Blanchard-Fields, F., Mienaltowski, A. & Seay, R. B. Age differences in everyday problem-solving effectiveness: Older adults select more effective strategies for interpersonal problems. *The Journals of Gerontology Series B: Psychological Sciences and Social Sciences* **62**, P61-P64 (2007).
- 34 Blanchard-Fields, F., Stein, R. & Watson, T. L. Age differences in emotion-regulation strategies in handling everyday problems. *The Journals of Gerontology Series B: Psychological Sciences and Social Sciences* **59**, P261-P269 (2004).
- 35 Scheibe, S. & Blanchard-Fields, F. Effects of regulating emotions on cognitive

- performance: what is costly for young adults is not so costly for older adults. *Psychology and aging* **24**, 217 (2009).
- 36 Birditt, K. S., Fingerman, K. L. & Almeida, D. M. Age differences in exposure and reactions to interpersonal tensions: A daily diary study. *Psychology and aging* **20**, 330 (2005).
- 37 Charles, S. T. & Piazza, J. R. Age differences in affective well-being: Context matters. *Social and Personality Psychology Compass* **3**, 711-724 (2009).
- 38 Blanchard-Fields, F. & Coats, A. H. The experience of anger and sadness in everyday problems impacts age differences in emotion regulation. *Developmental psychology* **44**, 1547 (2008).
- 39 Blanchard-Fields, F., Jahnke, H. C. & Camp, C. Age differences in problem-solving style: The role of emotional salience. *Psychology and aging* **10**, 173 (1995).
- 40 Birditt, K. S. & Fingerman, K. L. Age and gender differences in adults' descriptions of emotional reactions to interpersonal problems. *The Journals of Gerontology Series B: Psychological Sciences and Social Sciences* **58**, P237-P245 (2003).
- 41 Carstensen, L. L., Isaacowitz, D. M. & Charles, S. T. Taking time seriously: A theory of socioemotional selectivity. *American psychologist* **54**, 165 (1999).